EMBO
Molecular Medicine

# Identification and characterization of GLDC as host susceptibility gene to severe influenza

Jie Zhou[1,2,3,†], Dong Wang[2,†], Bosco Ho-Yin Wong[2], Cun Li[2], Vincent Kwok-Man Poon[2], Lei Wen[2], Xiaoyu Zhao[2], Man Chun Chiu[2], Xiaojuan Liu[2], Ziwei Ye[2], Shuofeng Yuan[2], Kong-Hung Sze[2], Jasper Fuk-Woo Chan[1,2,3,4,5,6], Hin Chu[1,2,3], Kelvin Kai-Wang To[1,2,3,4,5,6] & Kwok Yung Yuen[1,2,3,4,5,6,*] [iD]

## Abstract

Glycine decarboxylase (GLDC) was prioritized as a candidate susceptibility gene to severe influenza in humans. The higher expression of GLDC derived from genetic variations may confer a higher risk to H7N9 and severe H1N1 infection. We sought to characterize GLDC as functional susceptibility gene that GLDC may intrinsically regulate antiviral response, thereby impacting viral replication and disease outcome. We demonstrated that GLDC inhibitor AOAA and siRNA depletion boosted IFNβ- and IFN-stimulated genes (ISGs) in combination with PolyI:C stimulation. GLDC inhibition and depletion significantly amplified antiviral response of type I IFNs and ISGs upon viral infection and suppressed the replication of H1N1 and H7N9 viruses. Consistently, GLDC overexpression significantly promoted viral replication due to the attenuated antiviral responses. Moreover, GLDC inhibition in H1N1-infected BALB/c mice recapitulated the amplified antiviral response and suppressed viral growth. AOAA provided potent protection to the infected mice from lethal infection, comparable to a standard antiviral against influenza viruses. Collectively, GLDC regulates cellular antiviral response and orchestrates viral growth. GLDC is a functional susceptibility gene to severe influenza in humans.

**Keywords** antiviral response; genetic susceptibility; GLDC; severe influenza; viral replication

**Subject Categories** Immunology; Microbiology, Virology & Host Pathogen Interaction; Pharmacology & Drug Discovery

## Introduction

Human influenza viruses, such as seasonal influenza viruses and pandemic 2009 A(H1N1) virus (H1N1 hereinafter), usually lead to upper respiratory infection with mild-to-moderate symptoms (To et al, 2013), albeit occasionally causing life-threatening pneumonia in patients with chronic underlying diseases, or even in healthy individuals. A novel avian A(H7N9) virus has caused recurrent outbreaks of human infections in China since 2013 (Chen et al, 2013). Human H7N9 infection, mostly manifested as rapidly progressive pneumonia, showed variable susceptibility among individuals (Wang et al, 2017). The contribution of human genetic variations to susceptibility to severe influenza was clearly demonstrated in an earlier epidemiological study (Albright et al, 2008). The variable susceptibility to H7N9 infection and diversified disease severity in pandemic H1N1 infection prompted us to decipher the genetic susceptibility and identify host gene predisposing to severe influenza, i.e., H7N9 and severe H1N1 infection.

To identify host susceptibility gene to severe influenza, we previously conducted genome-wide association studies (GWAS) in two study cohorts: the H7N9 cohort including H7N9 patients versus healthy controls who were heavily exposed to the virus (Chen et al, 2015), and the H1N1 cohort consisting of severe H1N1 patients versus control patients with mild H1N1 infection (Zhou et al, 2012). To prioritize the candidate susceptibility genes to H7N9 and severe H1N1 infection, we integrated the GWAS discovery with expression quantitative trait loci (eQTL) analysis, a strategy which has been widely used to identify the trait gene targets (Musunuru et al, 2010; Zhu et al, 2016). Glycine decarboxylase (GLDC) emerged as a potential susceptibility gene in both study cohorts.

GLDC, the P protein of glycine cleavage system, binds to glycine and enables the transfer of methylamine group of glycine to T protein. Interestingly, Zhang et al uncovered a novel role of GLDC, which can promote pyrimidine biosynthesis and drive tumorigenesis of lung cancer (Zhang et al, 2012). It has been well-established that cellular pyrimidine metabolism profoundly affects virus propagation. Dihydroorotate dehydrogenase (DHODH), an enzyme catalyzing the fourth step of de novo pyrimidine biosynthesis, was repetitively identified as the cellular target in high-throughput

1 State Key Laboratory of Emerging Infectious Diseases, The University of Hong Kong, Pokfulam, Hong Kong
2 Department of Microbiology, The University of Hong Kong, Pokfulam, Hong Kong
3 Research Centre of Infection and Immunology, The University of Hong Kong, Pokfulam, Hong Kong
4 Carol Yu Centre for Infection, The University of Hong Kong, Pokfulam, Hong Kong
5 The Collaborative Innovation Center for Diagnosis and Treatment of Infectious Diseases, The University of Hong Kong, Pokfulam, Hong Kong
6 Department of Clinical Microbiology and Infection Control, The University of Hong Kong-Shenzhen Hospital, Shenzhen, China
*Corresponding author. Tel: +852 22554892; Fax: +852 28551241; E-mail: kyyuen@hku.hk
†These authors contributed equally to this work

screening for antivirals conducted by several groups, including us (Hoffmann *et al*, 2011; Wang *et al*, 2011; Cheung *et al*, 2017). Depletion of cellular pyrimidine inhibits the replication of an array of RNA viruses, DNA viruses, and retroviruses. Interestingly, Lucas-Hourani *et al* (2013, 2017) reported that pyrimidine deprivation suppresses viral growth through amplifying innate immune response, rather than depleting nucleotide precursors for viral genome replication as assumed previously (Hoffmann *et al*, 2011). Consolidating prior knowledge and the findings of genetic association studies, we hypothesized a link between GLDC and innate immunity, which may impact viral growth and underlie the genetic association of GLDC with H7N9 and severe H1N1 infection. We performed a series of *in vitro* and *in vivo* experiments and established GLDC as a functional susceptibility gene to severe influenza.

## Results

### GLDC was prioritized as a susceptibility gene to H7N9 infection and severe H1N1 infection by integrating GWAS and eQTL analysis

In order to identify the host gene(s) predisposing to severe pandemic H1N1 influenza, we conducted a small-scale GWAS in severe H1N1 patients and control patients with mild disease. An intronic SNP of GLDC, rs1755609, was significantly associated with the susceptibility to severe H1N1 infection. We verified the genetic association in a larger cohort including 174 severe patients and 258 mild controls. The carriers of AA and AG genotypes exhibited a 1.65-fold higher risk to severe H1N1 infection than the carriers of genotype GG (Table 1). We chose rs1755609 for validation since this SNP is an expression quantitative trait loci (eQTL) in 77 and 72 lymphoblast cell lines (LCLs) derived from Chinese and Japanese, respectively (Fig 1A), and an eQTL in human lung tissues (Fig 1B; Hao *et al*, 2012). Specifically, the differential expression levels of GLDC are significantly correlated to rs1755609 genotypes in LCLs and human lung tissues, the risk variants corresponding to higher GLDC expression. Interestingly, the genetic association of GLDC with severe H1N1 influenza was replicated in the H7N9 cohort including 102 H7N9 patients and 106 controls of healthy poultry workers. rs2438409, a SNP in high linkage equilibrium with rs1755609 in multiple ethnic groups (Fig 1C), was significantly associated with the susceptibility to human H7N9 infection (Table 2). The risk allele G, in high linkage with rs1755609 A allele, was significantly over-represented in H7N9 patients than in controls. The carriers of GG and GA genotypes were conferred more than twofold higher risk to H7N9 infection compared to carriers of genotype AA. Collectively, the integration of disease association and eQTL datasets suggested that higher GLDC expression encoded by genetic variations may confer predisposition to H7N9 infection and severe H1N1 infection.

### GLDC inhibition suppressed pyrimidine synthesis and boosted antiviral response

To substantiate GLDC as a functional susceptibility gene, we searched for the molecular mechanism(s) underlying the genetic association. Consolidating the existing knowledge, we hypothesized

**Table 1. Genetic association of rs1755609 with severe H1N1 infection.**

| Distribution and analysis | Severe patients (*n* = 174) | Mild controls (*n* = 258) |
|---|---|---|
| Genotype distribution, *n* (%) | | |
| AA | 52 (29.9) | 56 (21.7) |
| AG | 87 (50.0) | 126 (48.8) |
| GG | 35 (20.1) | 76 (29.5) |
| Allelic analysis (allele A) | | |
| OR | 1.421 | |
| *P* | 0.0115 | |
| Genotype analysis (dominant model) | | |
| OR | 1.653 | |
| *P* | 0.0292 | |

OR, odds ratio.

a link between GLDC and innate immunity. GLDC probably acts as an intrinsic regulator of host antiviral response due to its role in the *de novo* pyrimidine biosynthesis (Zhang *et al*, 2012), thereby impacting viral growth and disease outcome upon virus exposure. To establish the link that GLDC can modulate pyrimidine biosynthesis and regulate antiviral response, we performed two sets of experiments. Firstly, we intended to verify the role of GLDC in cellular pyrimidine biosynthesis as shown previously (Zhang *et al*, 2012). After treatment with a GLDC inhibitor (aminooxy)acetic acid (AOAA) in A549 cells for 48 h, the amount of thymidine, one of pyrimidine compounds, was detected with LC-MS/MS. The GLDC inhibitor AOAA appeared to diminish the cellular pool of thymidine nucleoside in a dose-dependent manner (Fig EV1), which verified the role of GLDC as an intrinsic regulator of pyrimidine biosynthesis as reported previously.

Secondly, we assessed whether GLDC inhibition can boost innate antiviral response. To this end, A549 cells transfected with IFNβ luciferase reporter plasmid were treated with AOAA or mock-treated for 24 h and then were applied to luciferase assay. However, AOAA treatment was unable to activate IFNβ luciferase reporter gene (Fig 2A, 0 μg/ml Poly I:C). We postulated that AOAA may require additional signal to trigger IFN activation, similar to the pyrimidine inhibitor in a previous study (Lucas-Hourani *et al*, 2013). To set up such a scenario, after transfection of the IFNβ reporter plasmid, Poly I:C, an agonist of pattern recognition receptors (PRR) TLR3 and RIG-I, was subsequently transfected along with AOAA treatment. After Poly I:C transfection, IFNβ reporter gene was slightly activated. Notably, in combination with Poly I:C stimulation, AOAA treatment further amplified IFNβ activation to a significantly higher level than the mock treatment. We further evaluated the effect of GLDC inhibition in an A549 cell line (A549-Dual) stably carrying an interferon-stimulated response element (ISRE) reporter gene. ISRE is a conserved DNA sequence element harbored in the promoters of most interferon-stimulated genes (ISGs). Likewise, AOAA treatment alone was unable to activate ISRE reporter gene. However, with simultaneous Poly I:C stimulation, AOAA further boosted ISRE activation to a significantly higher level. A549-Dual KO-RIG-I cells were generated from A549-Dual cells through the stable knockout of RIG-I gene. In these cells, AOAA-amplified activation of ISRE reporter

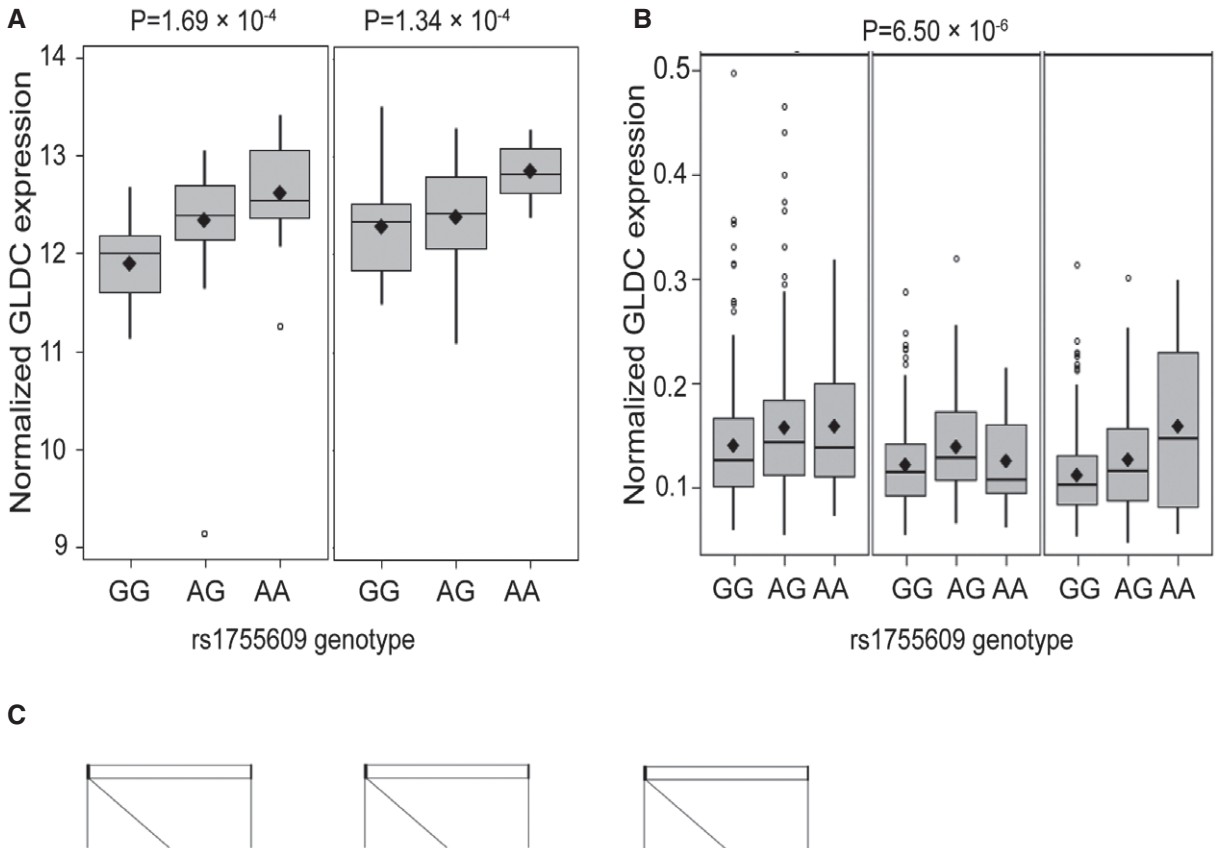

**Figure 1. Genotype-expression correlation of association SNP.**

A, B  The genotype-expression correlation of rs1755609 and GLDC in lymphoblast cell lines (LCLs, A) and human lung tissues (B). The box denotes the interquartile range; thick line within the box is the median; diamond represents the mean; whiskers are minimum and maximum and open dot is outlier. The genotype-expression correlation pattern in LCLs from 77 Chinese and 72 Japanese are plotted in the left and right, respectively. The lung eQTL data from three centers are presented. Linear regression is used for data analysis.

C  The high linkage disequilibrium (LD) pattern in three SNPs, including two association SNPs rs2438409 and rs1755609 (boxed), in multiple populations. The LD pattern is plotted using genotyping data of 77 Chinese Han in Beijing (CHB), 72 Japanese (JPT), and 73 Caucasian of European ancestry living in Utah USA (CEU) from 1000 Genomes Project. The boxes are colored according to $r^2$ measure on a white and gray scale. The numbers inside the boxes are $r^2$ measure. The gray box without number indicates the highest $r^2$ of 1.0.

gene was abolished (Fig 2A), indicating the requirement of RIG-I/PRR signaling for ISRE activation.

To ascertain the role of GLDC for modulating type I IFN production, we depleted GLDC with GLDC siRNA, followed with Poly I:C transfection. At 24 h after Poly I:C stimulation, cells and cell-free culture media were harvested. As shown in Fig 2B, GDLC expression decreased to around 20% in the GLDC-depleted cells compared with the scrambled siRNA-transfected cells. In the GLDC-depleted cells, IFNα mRNA expression was significantly upregulated. Consistently, IFNα secretion into the culture media was also elevated significantly. Together, we demonstrated that GLDC inhibition could suppress pyrimidine biosynthesis. In combination with PRR stimulation, GLDC inhibition and depletion amplified the induction of type I IFNs and ISGs. In addition, GLDC inhibition triggered activation of type I IFNs and ISGs required PRR signaling.

**Table 2.  Genetic association of rs2438409 with H7N9 infection.**

| Distribution and analysis | H7N9 patients (*n* = 106) | Healthy controls (*n* = 102) |
|---|---|---|
| Genotype distribution, *n* (%) | | |
| GG | 34 (32.1) | 19 (18.6) |
| GA | 51 (48.1) | 48 (47.1) |
| AA | 21 (19.8) | 35 (34.3) |
| Allelic analysis (allele G) | | |
| OR | 1.776 | |
| *P* | 0.0043 | |
| Genotype analysis (dominant model) | | |
| OR | 2.215 | |
| *P* | 0.0184 | |

OR, odds ratio.

## GLDC inhibition and depletion suppressed replication of influenza viruses via boosting antiviral response

We then evaluated whether GLDC inhibition can affect the replication of influenza viruses now that the inhibition boosted type I IFNs and ISGs. As shown in Fig 3, AOAA treatment significantly decreased the viral loads in both cell lysates and cell-free culture media (supernatant) in H1N1 virus- and H7N9 virus-infected A549 cells. AOAA treatment significantly reduced viral titer by more than 1 log unit in the H1N1-infected cells at 24 h post-infection (hpi, Fig 3A). The AOAA-mediated suppression of viral replication was more prominent in the H7N9-infected cells; treatment of 50 μM AOAA significantly decreased viral titer around 2 log units at 48 hpi (Fig 3B). As expected, AOAA-amplified antiviral response contributed to the decreased viral replication. In the H1N1-infected cells, AOAA treatment significantly augmented the production of type I IFNs (IFNα and IFNβ) and some ISGs in a dose-dependent pattern (Fig 4A). In accordance with the stronger suppression of viral growth, the boosted antiviral response was more pronounced in the H7N9-infected cells. At 24 hpi, AOAA treatment triggered a more intensive and extensive upregulation of type I IFNs and most ISGs (including ISG15, OAS1, PKR, IFI6, IFI27, IFI35, and IFIT3) in a dose-dependent manner (Fig 4B). These antiviral molecules also showed the consistent upregulation profile at 48 hpi (Fig EV2A). We also performed a cell viability assay to exclude the possibility that AOAA-mediated suppression of viral growth is related to any cytotoxicity (Fig EV2B).

As aforementioned, DHODH is an important enzyme in *de novo* pyrimidine biosynthesis pathway. Brequinar, a well-known inhibitor of DHODH (Liu *et al*, 2000), was shown to amplify the expression of ISGs (Lucas-Hourani *et al*, 2013). We hypothesize that brequinar could recapitulate the effects of AOAA in H7N9 viral replication if these two compounds truly target the enzymes in pyrimidine biosynthesis pathway. As expected, brequinar treatment prompted a heightened induction of IFNα and ISGs and significantly suppressed H7N9 replication (Fig EV3). The results lend further support to our finding that antiviral effect of AOAA is attributed to the pyrimidine deprivation mediated by GLDC inhibition.

To verify the role of GLDC on viral replication, multicycle replication of H7N9 virus was performed in A549 cells after GLDC siRNA depletion. The intracellular (cell lysate) and extracellular (supernatant) viral loads were significantly lower in GLDC siRNA-transfected cells than in control siRNA-transfected cells at 48 hpi (Fig 5A). Viral titer in the GLDC-depleted cells was also significantly lower than that in control cells. Furthermore, the amplified antiviral response observed in the AOAA treatment was replicated in the GLDC-depleted cells. The expression levels of IFNα and IFNβ were significantly augmented in the GLDC-depleted cells than in the control cells (Fig 5B). Several ISGs (e.g., IFIT1, IFIT3, IFIT35, and ISG15) were significantly upregulated in the former. Collectively, GLDC inhibition and depletion significantly suppressed the replication of H1N1 and H7N9 virus via the amplified antiviral response.

## GLDC overexpression promoted replication of influenza virus via the compromised antiviral response

To further establish the role of GLDC in replication of influenza viruses, we transfected GLDC expression plasmid or the vector and inoculated H1N1 or H7N9 virus 24 h post-transfection. GLDC overexpression significantly increased the viral loads in H7N9-infected cells (Fig 5C). The intracellular viral load was significantly higher in the GLDC-overexpressed cells than in the control cells at 24 hpi. At 48 hpi, the viral load in the culture media was more than 20-fold higher in the former than in the latter. Consistently, a significantly attenuated antiviral response, including type I IFNs and ISGs, was observed in the GLDC-overexpressed cells compared to the control cells (Fig 5D). The enhanced viral replication and blunted antiviral response were also observed in H1N1-infected cells after GLDC overexpression (Fig EV4). Therefore, GLDC overexpression compromised the antiviral response and promoted viral replication. The efficiency of GLDC siRNA depletion and GLDC overexpression is illustrated in Fig 5E.

Since GLDC affected growth of influenza viruses via modulating antiviral response, we tested whether this is operational in other viral infections. To this end, A549 cells were inoculated with Middle East respiratory syndrome coronavirus (MERS-CoV), an emerging virus causing human respiratory infection (Zhou *et al*, 2014), at 24 h post-transfection of GLDC expression plasmid or vector. We demonstrated that GLDC overexpression significantly promoted the replication of MERS-CoV at 48 hpi (Fig 5F), indicating that GLDC-orchestrated viral growth was operational in other viral infections.

## GLDC inhibition reduced the viral loads and protected the mice from lethal infection of H1N1 virus

We then assessed whether GLDC inhibition in the H1N1-infected BALB/c mice can recapitulate the amplification of antiviral response and suppression of viral growth and whether GLDC inhibition can protect the infected mice from severe outcome. Three groups of BALB/c mice were intranasally inoculated with the mouse-adapted pandemic 2009 H1N1 virus and intranasally administrated with AOAA or zanamivir or PBS for three times as illustrated in Fig 6A. A total of 9 mice in each group were monitored for body weight change and survival rate for 14 days post-inoculation. In contrast to the survival rate of 11.1% in PBS-treated mice, all AOAA-treated mice survived the infection (Fig 6B). Notably, AOAA provided an equivalent protection as zanamivir, a standard antiviral against influenza viruses, indicating that temporal GLDC inhibition could achieve a potent protection from fatal infection. The PBS-treated mice lost body weight significantly from day 4 post-inoculation while the body

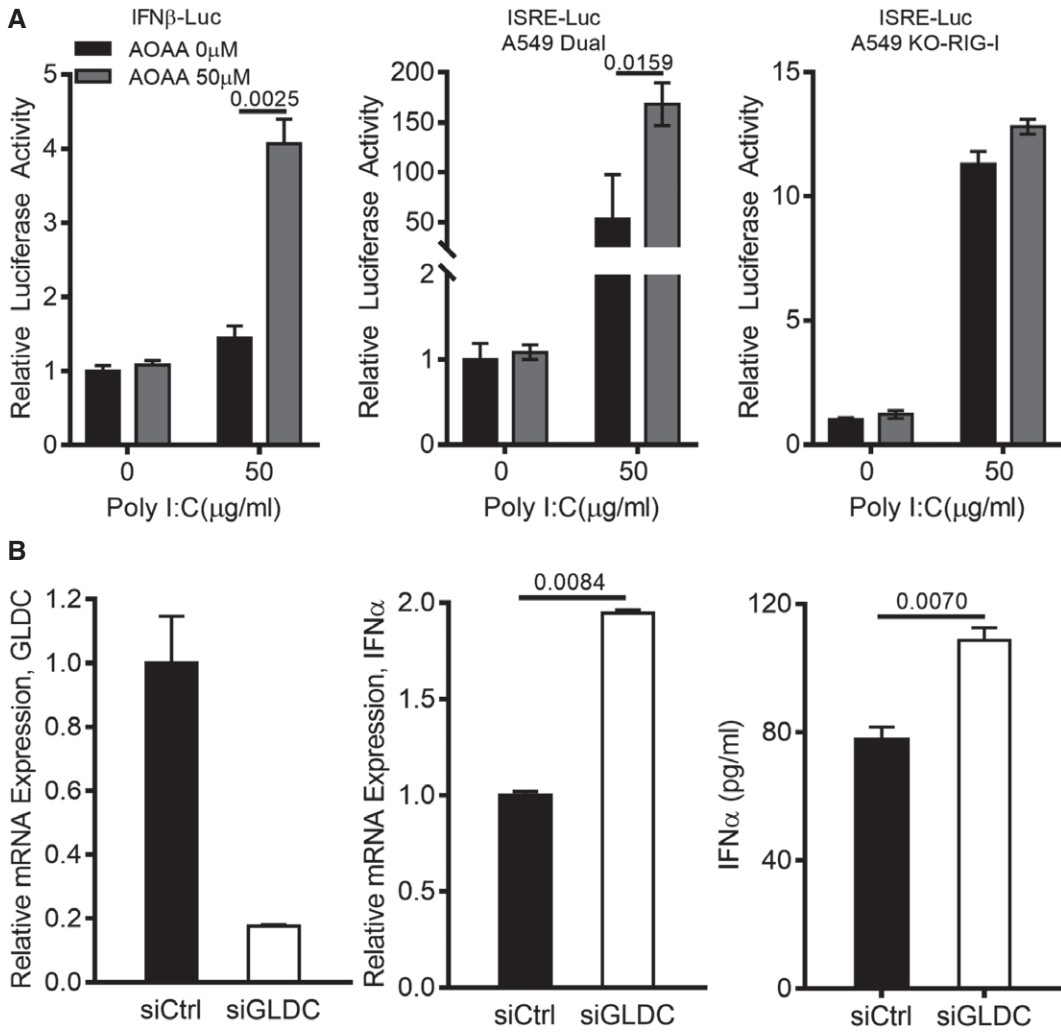

**Figure 2.  GLDC inhibition and depletion activated type I IFNs and ISGs.**

A    A549 cells co-transfected with pIFNβ-Luc and TK-Nano-Luc were further transfected with Poly I:C (0 and 50 μg/ml) and simultaneously treated with AOAA (0 and 50 μM). The cells were lysed for luciferase assay after 16 h. A549-Dual and A549-Dual KO-RIG-I cells carrying ISRE reporter gene were transfected and treated with the indicated concentrations of Poly I:C and AOAA respectively in triplicate. After 16 h, the cell-free supernatants were used for luciferase assay. The data present the normalized luciferase activity relative to mock Poly I:C transfection and mock AOAA treatment. Data shown are representative of three independent experiments, *n* = 3.

B    A549 cells transfected with GLDC siRNA (siGLDC) or scrambled siRNA (siCtrl) were further transfected with Poly I:C (50 μg/ml). After 24 h, the cell lysates and media were harvested for detection of mRNA expression of GLDC and IFNα by RT–qPCR, and measurement of IFNα production by ELISA, respectively. Data shown are representative of three independent experiments, *n* = 3.

Data information: Unpaired *t*-test is used for data analysis. Graphs show mean ± SD.

weights of AOAA-treated mice rebounded at days 3 and 4 (Fig EV5). Afterward, the body weights of AOAA-treated mice tended to be higher than those of the PBS-treated mice, yet comparable to those of zanamivir-treated mice. The viral load and viral titer in lung tissues of AOAA-treated mice were significantly lower than those in PBS-treated mice at days 3 and 5 post-infection (Fig 6C). In addition, a significantly higher level of IFNα was observed in the mouse lung tissues at days 3 and 5 post-infection in AOAA-treated mice than in PBS-treated mice (Fig 6D). The AOAA-amplified ISG production was also observed in the mice. The virus antigen-positive cells in mouse lung were more abundant in PBS-treated mice than in AOAA-treated mice (Fig 6E), which is consistent with the higher viral titers in the former. Again, we excluded the possibility that AOAA-mediated

reduction of viral growth was related to toxicity since the mice intranasally administrated with AOAA for 5 consecutive days did not exhibit any discernible difference in body weight loss in comparison with the mice treated with PBS (Fig EV5). Thus, GLDC inhibition in the H1N1-infected mice significantly boosted the antiviral response, decreased the viral growth, and protected the mice from lethal infection.

## Discussion

Influenza viruses cause mild-to-moderate respiratory illness in most patients, and occasionally severe or fatal infections. An earlier

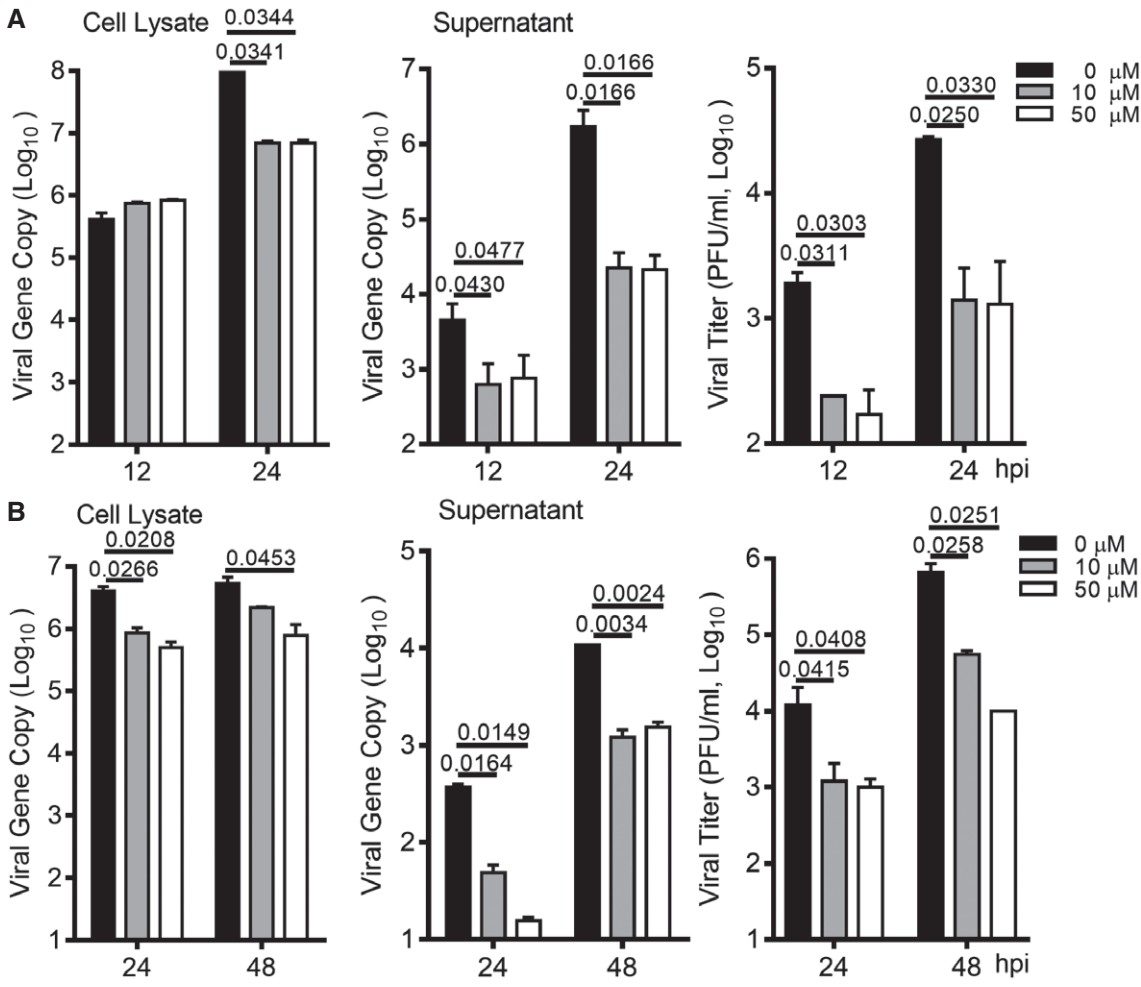

**Figure 3. GLDC inhibition suppressed viral replication.**

A, B A549 cells were inoculated with H1N1 (A) and H7N9 (B) viruses at an MOI of 0.01. The cells were treated with AOAA (0, 10, 50 μM) in triplicate prior to and after the inoculation. At the indicated hours post-inoculation, cell lysates and cell-free media (supernatant) were harvested for the detection of viral load with RT–qPCR assay, and viral titration with plaque assay, respectively. Data shown are representative of at least three independent experiments, $n = 3$. Graphs show mean ± SD. Unpaired $t$-test is used for data analysis.

epidemiological study clearly demonstrated the genetic contribution to fatal influenza infections (Albright *et al*, 2008). In addition, host genetic variations were found to be associated with the development of severe pneumonia in influenza infections (Zuniga *et al*, 2012). We sought to identify susceptibility gene(s) which may account for the inter-individual variability to severe influenza from the GWAS previously conducted in the H7N9 cohort and the pandemic H1N1 cohort. Besides the genetic contribution, emerging evidence suggested that broadly protective immune responses due to the childhood infection can provide long-term cross-immunity among different HA subtypes, especially those in the same phylogenetic group (Gostic *et al*, 2016).

At the first glance, the susceptibility to H7N9 infection and the susceptibility to severe H1N1 infection are not equivalent issues. In fact, H7N9 infection and severe H1N1 infection essentially shared the same pathology, i.e., viral pneumonia, the most lethal consequence of influenza (Nikolaidis *et al*, 2017). Overall, pandemic H1N1 viruses exhibit binding specificity to human-type α2,6-linked

sialic acid receptor (Zhou *et al*, 2013); the viruses tend to replicate more actively in human bronchus than in lung tissue (Chan *et al*, 2013). In nonhuman primates, the replication of pandemic H1N1 virus mainly affected the upper respiratory tract (Watanabe *et al*, 2013). On the other hand, H7N9 viruses bind to both the human-type α2,6-linked sialic acid receptor and the avian-type α2,3-linked receptor, which are mainly distributed in human airway epithelium and lower respiratory epithelium, respectively. Accordingly, H7N9 virus can infect epithelial cells in human airway and lower respiratory tract including alveoli (Knepper *et al*, 2013; Zhou *et al*, 2013) and replicated efficiently in both the upper and the lower respiratory tracts in nonhuman primates. Thus, the biological characteristics of pandemic H1N1 virus and H7N9 virus largely determine the predilection of pandemic H1N1 infection as an upper respiratory tract infection, and H7N9 virus more likely to develop a lower respiratory tract infection (Li *et al*, 2014). Given the existence of risk factors that facilitate viral growth or access the lower respiratory

 

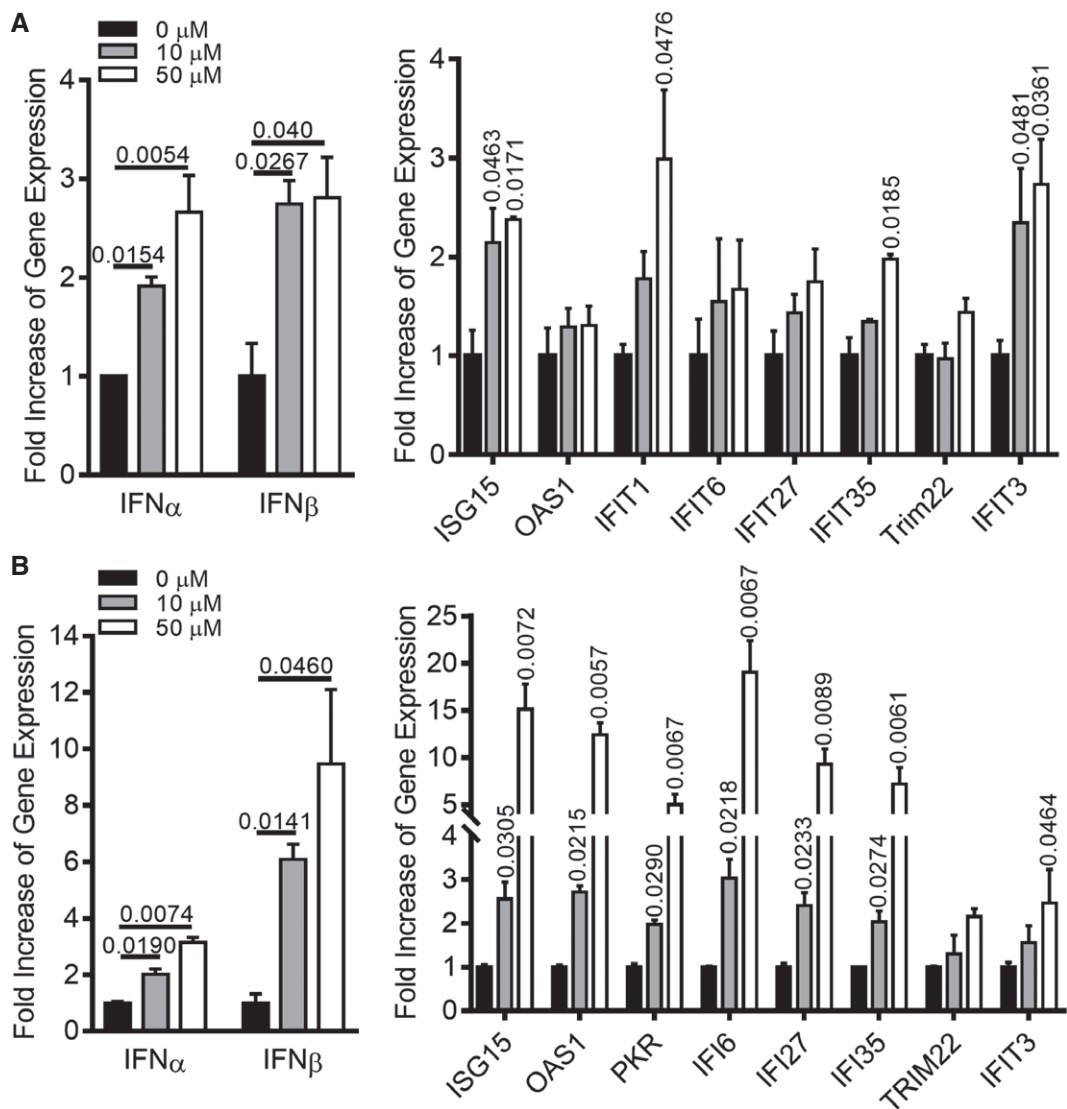

**Figure 4.  The amplified antiviral response contributed to AOAA-mediated inhibition of viral replication.**

A, B    A549 cells were inoculated with H1N1 (A) and H7N9 (B) viruses. The cells were treated with AOAA (0, 10, 50 μM) prior to and after the inoculation. At the indicated hours post-inoculation, cell lysates were harvested for the detection of expression level of antiviral molecules with RT–qPCR assay. Data shown are representative of at least three independent experiments, *n* = 3. Graphs show mean ± SD. Unpaired *t*-test is used for data analysis.

tract, the replicating H1N1 viruses in the upper respiratory tract may spread downward and cause viral pneumonia. Collectively, regardless of the virus subtype, both H7N9 infection and severe infection by pandemic H1N1 virus are essentially lower respiratory tract infections and/or pneumonia.

In the last decade, GWAS has evolved from identifying genetic variants associated with diseases into revealing the function of variants underlying disease associations. As an unbiased genome-wide interrogation of genetic variations in association with diseases or clinical traits, GWAS could provide novel biological insights into disease pathophysiology and uncover new molecules and pathways related to diseases. GWAS of disease susceptibility demonstrated that the majority of association signals involve non-coding single nucleotide variants, so-called regulatory variants

that modulate gene expression (Knight, 2014). With the recognition of regulatory polymorphism in regulating gene expression, genome-wide correlation of genetic variants with gene expression has been systematically interrogated and established in various human tissues, known as quantitative trait loci (eQTL) (Fairfax & Knight, 2014). eQTL can transform the GWAS association variants into candidate genes, which can be brought forward for functional validation experiments *in vitro* and/or *in vivo* (Zhu *et al*, 2016). In this study, the intersection of the genetic associations and eQTL datasets suggested that higher GLDC expression encoded by genetic variations may predispose the affected individuals to H7N9 and severe H1N1 infection.

To substantiate GLDC as a functional susceptibility gene to severe influenza, we extensively searched for the possible

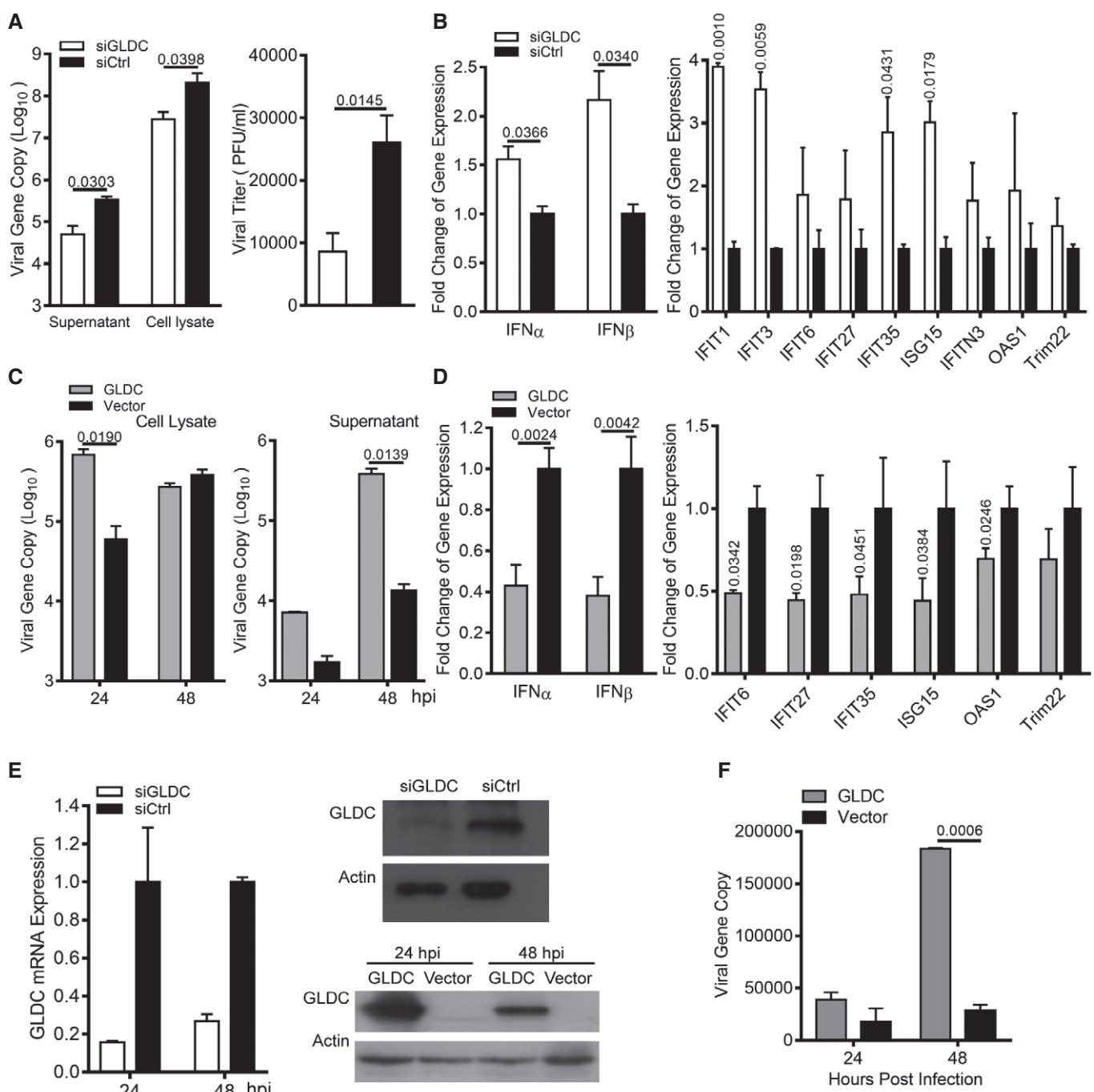

**Figure 5. GLDC depletion and overexpression affected viral replication through the altered antiviral response.**

A–D  A549 cells were transfected with GLDC siRNA (siGLDC) or scrambled siRNA (siCtrl) (A and B), or transfected with pcDNA3.1-GLDC or pcDNA3.1-His A vector (C and D). The transfected cells were inoculated with H7N9 virus at an MOI of 0.01. At 48 hpi, cell lysates and culture media (supernatant) of the siRNA-transfected cells were harvested for detection of viral loads and viral titers (A). The cell lysates were collected at 24 hpi to measure the expression levels of antiviral genes (B). At the indicated hpi, cell lysates and supernatants of the plasmid-transfected cells were harvested for viral load detection (C). The cell lysates collected at 24 hpi were used to measure the expression levels of antiviral genes (D). Data shown are representative of three experiments, *n* = 3. Graphs show mean ± SD. Unpaired *t*-test is used for data analysis.

E  The effective depletion of GLDC by siRNA was shown by RT–qPCR assay (left panel) and Western blot (48 hpi, upper right panel). GLDC overexpression at the indicated hpi was verified by Western blot (lower right panel). Data shown in left panel are representative of at least three experiments, *n* = 3. Graphs show mean ± SD. Unpaired *t*-test is used for data analysis.

F  A549 cells transfected with GLDC plasmid or vector were inoculated with MERS-CoV in triplicate. Cell-free culture media were harvested for detection of viral load at the indicated time points. Data shown are representative of two independent experiments, *n* = 3. Graphs show mean ± SD. Unpaired *t*-test is used for data analysis.

Source data are available online for this figure.

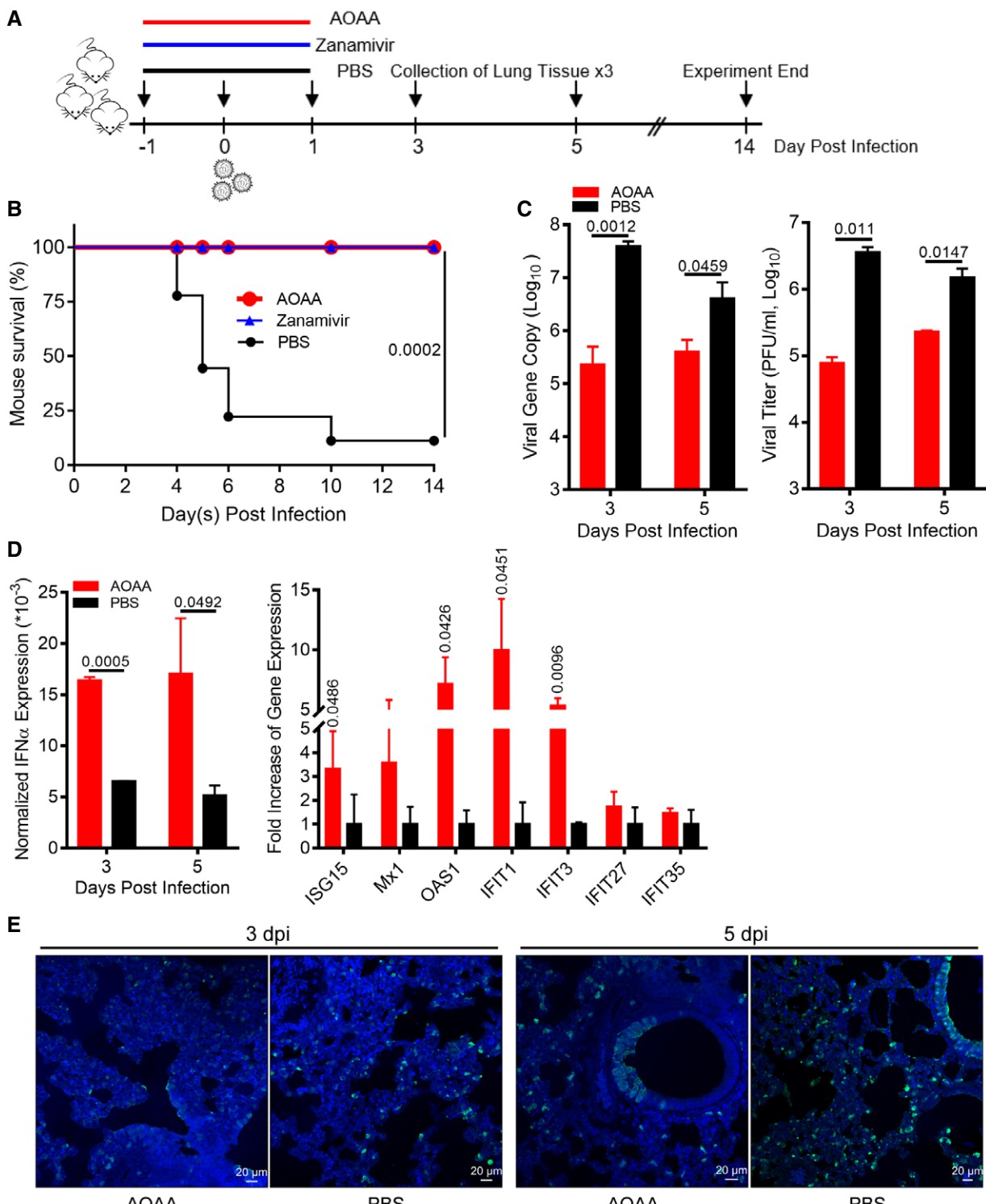

**Figure 6. GLDC inhibition reduced viral load and protected the mice from lethal infection of H1N1 virus.**

A    Regime of the mouse experiment.

B    Survival rates of AOAA-, zanamivir-, and PBS-treated mice after infection ($n = 9$). Mantel–Cox test is used for data analysis.

C    Viral loads and viral titers in the lung homogenates of the indicated mice ($n = 3$). Unpaired *t*-test is used for data analysis. Graphs show mean $\pm$ SD.

D    The normalized expression levels of IFNα in lung tissues of the indicated mice ($n = 3$) and fold change of ISGs at day 5 post-infection. Unpaired *t*-test is used for data analysis. Graphs show mean $\pm$ SD.

E    Representative images of virus-infected cells in the indicated mice. Mouse lung tissues are applied to immunofluorescence staining using the antibody against influenza virus NP (green) and confocal imaging.

mechanism(s). GLDC can modulate pyrimidine biosynthesis (Zhang *et al*, 2012); meanwhile, pyrimidine deprivation can suppress viral replication via boosting antiviral innate immunity (Lucas-Hourani *et al*, 2013). These evidences inspired us to hypothesize and subsequently uncover the GLDC/pyrimidine biosynthesis/innate immunity axis. We verify that a GLDC inhibitor AOAA substantially suppressed pyrimidine biosynthesis (Fig EV1). But AOAA alone was unable to activate IFNβ and ISRE luciferase reporter genes. In combination with TLR3/RIG-I stimulation, AOAA significantly activated both reporter genes (Fig 2A). Accordingly, the AOAA-amplified ISRE activation was RIG-I dependent since the activation was abolished in RIG-I-knockout cells. Thus, we uncovered a novel role of GLDC as an intrinsic regulator of innate immunity due to its effect in modulating pyrimidine biosynthesis. In addition, AOAA treatment and GLDC siRNA knockdown invariably suppressed the replication of H7N9 and H1N1 virus due to the amplified induction of type I IFNs and ISGs (Figs 3–5 and EV2). Consistently, GLDC overexpression promoted viral growth via the compromised antiviral responses (Figs 5 and EV4). These results established GLDC as an important host gene for the replication of influenza viruses due to the GLDC/pyrimidine biosynthesis/innate immunity axis. In addition, the axis may be functional in other viral infections since GLDC overexpression also significantly enhanced the replication of MERS-CoV (Fig 5F).

We further demonstrated that in the H1N1-infected BALB/c mice, AOAA-mediated amplification of antiviral response and suppression of viral growth (Fig 6) are even more prominent than its *in vitro* effect in A549 cells. The outperformance of AOAA *in vivo,* we believe, was attributed to AOAA-triggered antiviral cascade. As aforementioned, AOAA-amplified IFN and ISG response is only elicited in the context of virus infection or simultaneous stimulation of PRR. However, the cytopathic effect caused by replicating viruses would gradually compromise cell viability and cellular antiviral response. Thus, in the virus-infected A549 cells, AOAA has a limited window to boost antiviral response before the cells are overridden by replicating viruses. However, the infected cells only represented a small percentage in mouse airway, large amounts of uninfected healthy cells can benefit from the AOAA-amplified antiviral response. Thus, AOAA-amplified antiviral response was more favorably exemplified in the infected mice than in the infected A549 cells. Conceivably, the differential GLDC expression encoded by genetic variation, probably with a modest effect, could be magnified in the virus-infected individuals and substantially affect the susceptibility to severe influenza.

Influenza viruses are a family of pathogens that pose significant threat to public health globally. Numerous systemic or targeted screens have been performed *in vitro* to identify important host genes required for viral infection, replication, or pathogenesis (Karlas *et al*, 2010; Dittmann *et al*, 2015). However, the meta-analysis found little overlap among the host factors identified in various studies (Chou *et al*, 2015). Meanwhile, host factors essential for the pathogenesis of influenza have been extensively investigated in mouse studies. As the best characterized host restriction factor against influenza viruses *in vitro* and in mouse studies (Brass *et al*, 2009; Yount *et al*, 2010; Everitt *et al*, 2012), the importance of IFITM3 aroused the interest to explore its genetic association with severe influenza in humans. However, the results from various studies were controversial (Everitt *et al*, 2012; Mills *et al*, 2014).

From these lessons, we learned that the host genes identified from cellular and mouse studies may not be readily extrapolated to a clinical implication that these genes can really affect the disease presentation in human influenza. The candidate susceptibility genes to severe influenza in humans should be selected from the findings of human studies. However, GWAS, especially those in infectious diseases, are notoriously confounded by many factors. In our case, the added hindrance was the limited sample size due to the infrequent occurrence of severe influenza. Despite all the obstacles, we reason that important host genes for the pathogenesis of severe influenza may survive the considerable variations in human genetic association study, if less stringent criteria (*P* value and odd ratio) are applied to filter association variants, and if the expression level of candidate gene is genetically correlated and thus can be captured by eQTL analysis. Under this line of reasoning, GLDC was prioritized as a candidate susceptibility gene to severe influenza, which prompted us to conduct *in vitro* and mouse studies and validated GLDC as a functional susceptibility gene to severe influenza.

In this study, we have used a GLDC inhibitor AOAA to demonstrate the role of GLDC for antiviral response, viral growth and mouse survival, followed by GLDC siRNA depletion and GLDC overexpression experiments to verify the AOAA-mediated effect *in vitro*. We exclude the possibility that AOAA-mediated antiviral effect is related to any cytotoxicity *in vitro* and *in vivo*. In addition, in the H7N9-infected A549 cells, AOAA acts similarly as an inhibitor of DHODH, an essential enzyme catalyzing pyrimidine biosynthesis. AOAA, a commonly used GLDC inhibitor, is a substrate analogue of GLDC with the amino (NH2) group of glycine replaced by an aminooxy (ONH2) group. AOAA occupies the GLDC substrate-binding site and inhibits its activity (Nakai *et al*, 2005). However, the less prominent effect mediated by the genetic modifications (siRNA depletion and overexpression) than AOAA treatment calls for more clarification of the specificity of AOAA in future study. In addition, our data coherently reached a conclusion that pyrimidine deprivation mediated by GLDC inhibition/depletion could amplify the antiviral response of type I IFN and ISGs and suppress viral replication, which reinforced the link between pyrimidine biosynthesis pathway and antiviral status as consistently demonstrated in other studies (Hoffmann *et al*, 2011; Wang *et al*, 2011; Cheung *et al*, 2017). More recently, a pyrimidine inhibitor identified as a broad-spectrum antiviral was found to activate RIG-I (Chung *et al*, 2016), an essential cellular sensor to mount innate immune response in influenza virus infection (Loo *et al*, 2008). However, the detailed mechanism linking pyrimidine deprivation and the boosted antiviral response definitely requires further study.

As a genome-wide interrogation of genetic variations with human diseases/traits, GWAS have successfully identified novel or unexpected pathways and molecules that are involved in disease processes and can inform potential therapeutic targets (Okada *et al*, 2014; Smemo *et al*, 2014). However, to our knowledge, few GWAS discovery in infectious diseases has been deliberately corroborated to define a novel disease-related gene. In this study, we have utilized integrative approaches to identify and characterize GLDC as an important susceptibility gene to severe influenza. Our findings will not only reveal the biological pathway leading to severe influenza, but also be important for the clinical management of severe patients.

# Materials and Methods

### Cells, viruses, and plasmid

A549 cells (purchased from ATCC) were cultured in Dulbecco's modified Eagle's medium (DMEM) (Gibco) supplemented with 10% fetal bovine serum (FBS) (Gibco) and 100 units/ml penicillin and streptomycin at 37°C with 5% $CO_2$. A/Hong Kong/415742/2009 (H1N1)pdm09 and A/Anhui/1/2013 (H7N9) were used *in vitro* experiments; a mouse-adapted strain A/Hong Kong/415742Md/2009(H1N1)pdm09 was propagated in embryonated eggs and utilized for mouse experiments. The full-length cDNA of GLDC was amplified using the primer pair, GGGGTACCTATGCAGTC CTGTGCCAGGGCGT (forward) and GGTCTAGACTAAGAAGACG CCCTCTTTTG (reverse), then inserted into pcDNA3.1-His A vector with KpnI-HF and XbaI. All experiments with live viruses were conducted in biosafety level 2 or 3 laboratories upon the institutional approval.

### Luciferase reporter assay and detection of IFN production

After overnight culture in 24-well plate, A549 cells were co-transfected with 1,000 ng pIFNβ-Luc (Siu *et al*, 2009) and 1 ng TK-Nano-Luc (Promega) per well. At 24 h post-transfection, cells were further transfected with Poly I:C (0 or 50 μg/ml, InvivoGen) and were simultaneously treated with the indicated concentrations of AOAA (Sigma-Aldrich, CAS No. 2921-14-4) and incubated at 37°C for 16 h. Subsequently, the cells were lysed for luciferase assay using Dual-Glo Luciferase Assay System (Promega) in a Victor X3 Multilabel reader (PerkinElmer).

A549-Dual™ Cells and A549-Dual™ KO-RIG-I Cells (InvivoGen) were transfected with Poly I:C and treated with the indicated concentrations of AOAA for 16 h. Cell-free media were used for luciferase assay using QUANTI-Luc™ (InvivoGen). A549 cells were transfected with GLDC siRNA or scrambled siRNA (50 nM, Thermo Fisher Scientific) using Lipofectamine RNAiMAX (Thermo Fisher Scientific). Two days after siRNA transfection, the cells were further transfected with Poly I:C of 50 μg/ml and maintained in DMEM for 24 h. The cells and cell-free media were then harvested for detection of mRNA expression of GLDC and IFNα by RT–qPCR assay and measurement of IFNα production by VeriKine™ Human IFN Alpha ELISA Kit (PBL Assay Science), respectively.

### Virus infection

A549 cells in 24-well plate were either transfected with GLDC siRNA or scrambled siRNA as mentioned above and maintained for 2 days, or transfected with 500 ng pcDNA3.1-GLDC or pcDNA3.1-His A vector per well using Lipofectamine 3000 (Thermo Fisher Scientific) and maintained for 1 day, or pretreated with various concentrations of AOAA for 1 h. Subsequently, the transfected cells or pretreated cells were inoculated with H1N1 or H7N9 at a multiplicity of infection (MOI) of 0.01. The cells were maintained in DMEM containing 2 μg/ml TPCK-trypsin and 3% BSA. In AOAA experiments, various concentrations of AOAA were supplemented in the culture medium after inoculation. At the indicated time points, cell lysates and cell-free media were harvested for detection of viral load, cellular gene expression, and viral titration.

A549 cells were also inoculated with MERS-CoV at an MOI of 0.01 at 24 h after the cells were transfected with 1,000 ng of pcDNA3.1-GLDC or pcDNA3.1-His A vector per well. The infected cells were maintained in DMEM supplemented with 3% BSA. The cell-free culture media were harvested at the indicated time points for detection of viral load as we described elsewhere (Zhou *et al*, 2014).

### Mouse experiment

Female BALB/c mice of 6–8 weeks old were maintained in standard Biosafety level 2 animal laboratory in our institute and given access to standard pellet feed and water *ad libitum*. All animal experiments, housing, and husbandry followed the operating procedures approved by the Committee on the Use of Live Animals in Teaching and Research, the University of Hong Kong (project No. 4057-16). We follow the protocol to ensure that weight loss will not exceed 25% of mouse body weight. We intervene and prevent unnecessary suffering by euthanasia when an animal in such a state that survival is not possible. The regime of mouse experiment is illustrated in Fig 6A. The mice were randomly divided into three groups and inoculated with 20PFU of A/Hong Kong/415742Md/2009(H1N1)pdm09, a mouse-adapted strain of pandemic H1N1 virus (Zheng *et al*, 2010) after mice were anesthetized by intraperitoneal injection of ketamine/xylazine cocktail containing 70–100 mg/kg ketamine and 10–20 mg/kg xylazine. At 1 day prior to the inoculation, 12 h and 1 day post-inoculation, the mice were intranasally administered either with AOAA (10 mg/kg weight) or zanamivir (2.5 mg/kg weight) or PBS in a volume of 20 μl after anesthesia. The mice (*n* = 9) respectively treated with AOAA or zanamivir or PBS were monitored daily for body weight and survival for 14 days after inoculation. Three mice in AOAA-treated group and PBS-treated group were sacrificed at 3 and 5 days after viral challenge. Half of lung tissue was collected and homogenized for the quantification of viral growth and antiviral gene expression while the other half was fixed in 4% PFA for tissue processing and immunofluorescence staining.

### Quantification of viral load and mRNA expression level of cellular gene, viral titration

Cell lysates and homogenized mouse lung tissues were applied to RNA extraction, followed by reverse transcription using the virus-specific primer and oligo(dT). The resultant cDNAs were used for qPCR assay to measure viral load and expression level of cellular gene as described previously (Zhou *et al*, 2014). The cell-free media of infected cells and supernatants of homogenized mouse lung tissues were applied to viral RNA extraction with PureLink Viral RNA/DNA Mini Kit (Thermo Fisher Scientific) and RT–qPCR for the quantification of viral load by absolute quantification using a plasmid expressing a conserved region of IAV M gene. The qPCR primers are listed in Appendix Table S1. Viral titers in the culture media and supernatants of homogenized mouse tissues were detected with plaque assay in MDCK cells (Zhou *et al*, 2018).

### Western Blot and immunofluorescence staining

A549 cells transfected with plasmids or siRNAs were lysed in RIPA lysis buffer (Thermo Fisher Scientific) with protease inhibitor

**The paper explained**

**Problem**

The host genes essential for the pathogenesis of influenza viruses have been extensively investigated in cellular and mouse studies. However, most identified host genes are unable to be translated to a clinical implication that they can significantly affect the disease manifestation in influenza patients. Thus, it remains poorly understood which host genes may dictate the susceptibility to severe influenza in humans.

**Results**

GLDC was prioritized as a potential susceptibility gene to severe influenza through an integrative approach. Based on the existing knowledge, we uncovered the GLDC/pyrimidine biosynthesis/innate immunity axis. We demonstrated that GLDC inhibition and depletion significantly amplified the antiviral response of type I IFNs and IFN-stimulated genes and suppressed the replication of H1N1 and H7N9 viruses. Consistently, GLDC overexpression significantly promoted viral replication due to the attenuated antiviral responses. GLDC inhibition in the H1N1-infected BALB/c mice recapitulated the amplified antiviral response and suppressed viral growth, indicating that temporal GLDC inhibition could achieve a potent protection to the infected mice from the fatal infection.

**Impact**

We identified an important susceptibility gene to severe influenza. GLDC modulated antiviral response against influenza A viruses, thereby impacting viral growth and disease outcome.

cocktail (Roche) and applied to detect GLDC protein using anti-GLDC (Abcam ab97625, 1:1,000 dilution). Cellular β-actin was probed as loading control. Mouse lung slides were used for immunofluorescence staining with anti-nucleoprotein (NP, Millipore, MAB8257F-5, 1:1,000 dilution) to label the virus-infected cells as described elsewhere (Zhou *et al*, 2017) and imaged using Carl Zeiss LSM 800 confocal microscopy.

**Statistics**

For mice experiments, nine mice in each group for evaluating mouse survival and three biological triplicates at each time point for detecting the infection are the commonly used sample sizes which are appropriate for statistical analysis while minimizing animal use. The inbred and cohoused mice were randomly grouped, which minimized the subjective bias. No blinding was done. Mouse survival data were analyzed with Mantel–Cox test. In all the experiments, no sample was excluded for data analysis. Unpaired two-sided Student's *t*-test analysis was used for data analysis of all *in vitro* experiments which were performed at least three times. A $P < 0.05$ was considered to be statistically significant. Data are presented as mean and SD of representative experiments.

**Expanded View** for this article is available online.

**Acknowledgements**

This work was partly supported by the donations of the Shaw Foundation Hong Kong, Richard Yu and Carol Yu, Michael Seak-Kan Tong, Respiratory Viral Research Foundation Limited, Hui Ming, Hui Hoy & Chow Sin Lan Charity Fund Limited, and Chan Yin Chuen Memorial Charitable Foundation; and funding from Health and Medical Research Fund (HMRF, RRG-05 and 17161272) of Food and Health Bureau; the Collaborative Research Fund (C7011-15R) of the Research Grants Council, The Government of Hong Kong Special Administrative Region; the High Level Hospital-Summit Program in Guangdong, The University of Hong Kong-Shenzhen Hospital; Collaborative Innovation Center for Diagnosis and Treatment of Infectious Diseases, Ministry of Education of China; the National Project of Infectious Disease (2014ZX10004001004), Ministry of Science and Technology of China.

**Author contributions**

JZ and K-YY designed the study. JZ, DW, BH-YW, CL, XZ, MCC, and XL performed the *in vitro* experiments. ZY and K-HS performed LC-MS/MS analysis. DW, VK-MP, and SY conducted mouse experiments. JZ, DW, LW, HC, JF-WC, and KK-WT analyzed and interpreted the data. JZ, HC, JF-WC, KK-WT, and K-YY wrote and revised the manuscript.

**Conflict of interest**

The authors declare that they have no conflict of interest.

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
