## [Review Process File · EMBO Molecular Medicine]

Identification and characterization of GLDC as host susceptibility gene to severe influenza

Jie Zhou, Dong Wang, Bosco Ho-Yin Wong, Cun Li, Vincent Kwok-Man Poon, Lei Wen, Xiaoyu Zhao, Man Chun Chiu, Xiaojuan Liu, Ziwei Ye, Shuofeng Yuan, Kong-Hung Sze, Jasper Fuk-Woo Chan, Hin Chu, Kelvin Kai-Wang To, Kwok Yung Yuen

Review timeline:

Submission date:	11 July 2018
Editorial Decision:	14 August 2018
Revision received:	13 September 2018
Editorial Decision:	26 September 2018
Revision received:	3 November 2018
Accepted:	7 November 2018

Editor: Céline Carret

Transaction Report:

1st Editorial Decision

14 August 2018

Thank you for the submission of your manuscript to EMBO Molecular Medicine. We have now heard back from the three referees whom we asked to evaluate your manuscript. Although the referees find the study to be of potential interest, they also raise a number of concerns that must be thoroughly addressed.

You will see from the set of comments pasted below that the referees have some serious and partially relevant concerns and we would like to encourage you to focus on the following points: 1) the inhibitor used is not well described nor characterized and this should be done, 2) ethical endpoint/euthanasia selection criteria are ill-defined and should be described, 3) the weight loss/survival ratio is an issue and the three referees upon our cross-commenting exercise, emphasize the need to repeat the *in vivo*-mice experiment, 4) the translational relevance is unclear and hopefully points 1 and 3 above should help improve this last aspect that is critical for publication in EMBO Molecular Medicine.

We would welcome the submission of a revised version within three months for further consideration and would like to encourage you to address all the criticisms raised as suggested to improve conclusiveness and clarity. Please note that EMBO Molecular Medicine strongly supports a single round of revision and that, as acceptance or rejection of the manuscript will depend on another round of review, your responses should be as complete as possible.

I look forward to receiving your revised manuscript.

***** Reviewer's comments *****

Referee #1 (Remarks for Author):

Evidence presented in this study indicates that high expression of GLDC, a pyrimidine biosynthesis enzyme, increases human susceptibility to severe influenza virus infections via regulation of type I interferon induction. GLDC was identified as a possible susceptibility factor for severe H1N1 infection in a GWAS analyzing 174 severe patients and 258 mildly infected controls. This was further validated in an H7N9-infected patient cohort. The identified GLDC risk allele, rs1755609-A, correlates with increased GLDC expression when analyzing genotype/expression data from several previous studies. Mechanistically, the authors present loss of function and gain of function experiments showing that GLDC regulates the type I IFN response and replication of influenza virus in human cells. Finally, they observe that GLDC chemical inhibition in mice increases survival after H1N1 infection, correlating with decreased lung viral load and an increased IFN response. Overall, the data provided is convincing and supports the conclusion that GLDC represents a novel influenza virus susceptibility gene in cells and in vivo.

Minor comments:

1. I do not understand what the authors are attempting to show in Figure 1C.
2. Has the rs1755609-A been identified in the many other GWAS examining influenza susceptibility? If not, how is this study different?
3. The rs34481144 eQTL for IFITM3 that is also associated influenza susceptibility should be discussed. Was this risk allele also included in your GWAS? Might these risk alleles be additive or are they found in distinct patients?
4. Can the authors provide data or speculate as to how rs1755609 regulates GLDC expression?
5. Can the authors provide data or speculate as to how the pyrimidine biosynthesis pathway regulates type I IFN production? Are there other enzymes in this pathway that may be similarly involved?

Referee #2 (Comments on Novelty/Model System for Author):

The interpretation of the data relies heavily, though not entirely, on the use of an ill-defined inhibitor. The specificity of that inhibitor for GLDC should be documented. Reference to ethical approval of the mouse challenge experiment and the ethical endpoints that were used to minimize animal suffering appear to be missing.

Referee #2 (Remarks for Author):

The authors provide evidence based on an observational study in H1N1 and H7N9 patients with moderate or severe disease, that an intronic allele in the GLDC gene is associated with a more severe clinical outcome. Treatment of A549 cells with aminooxy-acetic acid combined with polyI,C transfection results in a higher type I IFN and ISG induction compared with control cells that were transfected with polyI,C only. Knock down of GLDC expression combined with polyI,C transfection had a similar effect. Treatment of A549 cells with AOAA reduced the replication of H1N1pdm and H7N9 infection, and was associated with increased type I IFN induction. Knock down of GLDC expression reduced H7N9 replication and overexpression of GLDC increased H7N9 replication in A549 cells. Finally, three consecutive intranasal AOAA treatments of mice partially protected the mice against disease following H1N1pdm challenge, which was associated with reduced virus replication and increased IFN α and ISG expression.

The link between the susceptibility to influenza and GLDC is novel and interesting. However, many of the results are based on the use of AOAA. The specificity and potential toxicity of this drug is poorly defined.

Major remarks:

1. Line 28. In contrast to the statement in the manuscript there is substantial evidence that pre-existing immunity has an important impact on the immune response and disease outcome following infection with pandemic H1N1 virus or H7N9 virus. This is presumably due to antigenic imprinting. See e.g. Gostic et al., *Science* 2016; reviewed in Cobey and Hensley, *COV* 2017). The statement should therefore be adapted.

2. Several susceptibility gene polymorphisms have been proposed to contribute to disease severity in influenza patients in the context of H1N1pdm or zoonotic infections with H7N9 viruses (IFITM3; Chen et al *Sci Rep.* 2016 May 9;6:25614. doi: 10.1038/srep25614.). The authors previously proposed that Galectin 1 was a susceptibility gene for H7N9. Now altered expression of GLDC is proposed as a susceptibility gene, mechanistically supposedly by its impact on pyrimidine biosynthesis, which in turn can suppress the innate antiviral response. There may thus be a more general effect on susceptibility to virus infection and GLDC expression or activation levels. The proposed link between GLCD and innate antiviral responses would be strengthened if the authors could extend the findings for another viral infection, for example VSV.

3. It is unclear how specific an inhibitor of GLDC AOAA is. Also, in EV1, a concentration of 100 microM does not seem to result in a significant reduction in thymidine in A549 cells. The statement in line 122 (GLDC inhibition could suppress pyrimidine biosynthesis) is therefore not supported by the data. The specificity of AOAA should be documented with GLDC knock down or -deficient cells before conclusions based on experiments with AOAA can be made. GLCD knock down cells were generated by the authors. The statement in line 125 (GLDC inhibition triggered activation of type I IFNs) is not correct. Without polyI,C transfection there appears to be no effect on type IFN or ISG induction.

4. Related to point 3: do DD264 or brequinar still have an anti-influenza virus effect in GLDC knock down cells?

5. The anti-H1N1 and H7N9 effect of AOAA seems clear in infected A459 cells as are the effects on IFN-alpha and -beta, and ISG expression. Please clarify that in these experiments AOAA was only applied to the cells before infection (pretreatment).

6. The mouse experiment documents limited protection of intranasal AOAA treatment of H1N1 pdm infected BALB/c mice. With 10 mice per group it is unclear what 12.5% survival means. In addition, the experiment should be repeated with H1N1 pdm or with H7N9 virus challenge. The authors should also provide data on the pharmacokinetics and possible toxicity of AOAA in vivo. GLDC deficiency in humans is associated with severe developmental defects.

Other remarks:

1. Panel 4E fits better in figure 2 where the GLDC KD cells are described.
2. GLCD expression levels are high in liver, kidney and placenta but low in the lung (Kure et al., *J Hum Genet.* 2001;46(7):378-84). How do the authors interpret their findings in the light of that report?
3. Line 254: "the outperformance...": compared to what? Certainly not to zanamivir administration.
4. Please specify whether the mice were sedated or not for the intranasal inoculations.
5. Line 433: antiviral should be replaced with anti-influenza A.

Referee #3 (Comments on Novelty/Model System for Author):

In the animal model, would the authors please explain the criteria for evaluating illness and death after infection that is presented in Figure 4B and EV4. As presented in EV4, the weight loss observed in the PBS group is moderate for an influenza virus infection model, with the average weight loss at about 12% and low points in the weight loss curves observed at days 4 and 9 after

infection. Do the authors have any explanation for the bimodal weight loss in these animals that demonstrates recovery at Day 6 followed by weight loss at Day 9? This is especially interesting when aligning the survival curves (Figure 4B) with weight loss because deaths occur on Days 5, 7, and 10 after infection. This observed weight loss can explain the deaths at Day 5 (with low weights at Day 4), but do not explain the deaths at Day 8 when weights return to nearly 100% at Day 7. The loss of weight at Day 9 corresponds with the death at Day 10. Two things are worth noting from the weight: survival comparison. First, the weight loss after infection is not consistent with the weight loss observed in other lethal models of influenza virus infection where a consistent progression toward 25-30% weight loss is observed. Second, a major loss of animals due to infection is associated with increases in group body weight (Day 7/8), while a minimal number of deaths is associated with a decrease in body weight (Day 9/10). If the authors could clarify how this weight loss is associated with the survival observed, and specifically comment on the euthanasia criteria for these animals, it would be greatly appreciated.

Referee #3 (Remarks for Author):

Comments for the authors of EMBO Molecular Medicine manuscript number EMM-2018-09528: The authors of the EMBO Molecular Medicine manuscript "Identification and characterization of GLDC as host susceptibility gene to severe influenza", present their method for demonstrating that glycine decarboxylase (GLDC) is associated with severe influenza infections. They begin this study by identifying GLDC using a genome-wide association study in patient populations that presented with severe infections caused by either H7N9 or pandemic H1N1 viruses. Genetic evaluation of these patients showed an association between severe a GLDC SNP and severe influenza infections. The authors then tested the contribution of GLDC to severe influenza infection by initially testing the response of influenza virus-infected cells to treatment with the GLDC inhibitor (aminooxy)acetic acid (AOAA) in A549 cells. Specifically, using a luciferase system, the authors show that type I IFN expression is increased in the presence of the GLDC inhibitor AOAA, indicating that GLDC limits anti-viral activity. These findings are corroborated using siRNA studies that show increased type I IFN expression when GLDC is depleted.

Having established an association between GLDC expression and reduced anti-viral immunity, modulated by type I IFN expression, the authors further show that AOAA treatment reduces viral infection of A549 cells, an effect that is again associated with increased expression of type I IFNs and subsets of IFN-stimulated genes (ISGs). Interestingly, the increase in ISGs is both more robust and broad after infection with H7N9 when compared with H1N1, indicating potential viral gene contributions to the phenotypes observed. The authors then use both siRNA depletion of GLDC and GLDC overexpression to show that GLDC expression directly affects virus replication, type I IFN expression, and ISG levels after infection. Finally, in a mouse infection model the authors show that AOAA is as effective as zanamivir at preventing death after influenza virus infection, an outcome that is associated with increased virus levels in lungs, IFN-alpha expression, ISG expression, and detection of virus nucleoprotein in the lung.

This manuscript does an excellent job of progressing the study from an initial characterization of a human gene associated with severe influenza virus infection to demonstrating that inhibition of this gene activity can directly improve outcomes after influenza virus infection. The authors show that the host type I IFN response, in particular ISGs, are important for the outcomes observed both through chemical and genetic inhibition of GLDC activity in cell lines and mice. This progression from a human gene target through demonstration that specific inhibition of gene function can directly influence the fate of an infection is a logical and complete progression of this gene from initial characterization to demonstrated gene function. While the information presented is of interest, there are a few issues with the presentation of the information and discussion of the findings that I would like the authors to consider as they evaluate the presentation of these findings.

General Comments:

1. One major concern with the data presented is the eventual utilization of a therapeutic that targets a gene product that needs to be inhibited in order to demonstrate an effect. Treatment of individuals with this product will need to overcome variations in gene expression within the human population that may greatly influence dosing and overall effectiveness. The most direct demonstration of effectiveness in a therapeutic setting is the murine study where AOAA is used to prevent death after infection, but the initial infection is still associated with weight loss and recovery. Furthermore, the

fact that AOAA is administered on days -1, 0, and +1 after infection brings into question the potential application of this therapeutic. How do the authors propose therapeutic treatment after influenza virus infection?

2. In the animal model, would the authors please explain the criteria for evaluating illness and death after infection that is presented in Figure 4B and EV4. As presented in EV4, the weight loss observed in the PBS group is moderate for an influenza virus infection model, with the average weight loss at about 12% and low points in the weight loss curves observed at days 4 and 9 after infection. Do the authors have any explanation for the bimodal weight loss in these animals that demonstrates recovery at Day 6 followed by weight loss at Day 9? This is especially interesting when aligning the survival curves (Figure 4B) with weight loss because deaths occur on Days 5, 7, and 10 after infection. This observed weight loss can explain the deaths at Day 5 (with low weights at Day 4), but do not explain the deaths at Day 8 when weights return to nearly 100% at Day 7. The loss of weight at Day 9 corresponds with the death at Day 10. Two things are worth noting from the weight: survival comparison. First, the weight loss after infection is not consistent with the weight loss observed in other lethal models of influenza virus infection where a consistent progression toward 25-30% weight loss is observed. Second, a major loss of animals due to infection is associated with increases in group body weight (Day 7/8), while a minimal number of deaths is associated with a decrease in body weight (Day 9/10). If the authors could clarify how this weight loss is associated with the survival observed, and specifically comment on the euthanasia criteria for these animals, it would be greatly appreciated.

3. Would the authors be willing to comment on the potential application of these findings and the therapeutic approach toward future treatments in humans. In particular, did the patients from which the genetic information was first derived have co-infections or secondary bacterial infections? How does GLDC influence secondary bacterial infections? Would AOAA therapy be expected to affect progression to a secondary bacterial infection? Would this effect be prevention or enhancement of this secondary infection?

4. The genetic observation made does not seem to associate disease severity with a loss of gene expression, a tool that was used to demonstrate the role of GLDC in the cell culture studies. In the individuals that have different SNPs for GLDC, are the levels of protein expression affected? What are the subsequent type I IFN responses in these individuals? Since these are type I IFNs, their baseline levels may be increased when compared to individuals with high levels of GLDC expression. Similarly, does ISG expression differ in these individuals? Finally, can AOAA be used clinically, and does it have any potential alternative effects that could be detrimental to the progression of a virus infection and/or a secondary bacterial infection?

5. Overall, the decrease in virus titers in the lungs of mice (Figure 4C) was approximately 1-2 logs with high levels still detected at Day 5 post-infection. Could the authors comment on the potential impact of a lingering virus infection after AOAA treatment.

Specific comments

1. While understandable at the level of data presentation and evaluation, the manuscript would benefit from substantial editing for sentence and paragraph structure throughout.

1st Revision - authors' response

13 September 2018

Referee #1 (Remarks for Author):

Evidence presented in this study indicates that high expression of GLDC, a pyrimidine biosynthesis enzyme, increases human susceptibility to severe influenza virus infections via regulation of type I interferon induction. GLDC was identified as a possible susceptibility factor for severe H1N1 infection in a GWAS analyzing 174 severe patients and 258 mildly infected controls. This was further validated in an H7N9-infected patient cohort. The identified GLDC risk allele, rs1755609-A, correlates with increased GLDC expression when analyzing genotype/expression data from several previous studies. Mechanistically, the authors present loss of function and gain of function experiments showing that GLDC regulates the type I IFN response and replication of influenza virus in human cells. Finally, they observe that GLDC chemical inhibition in mice increases survival after

H1N1 infection, correlating with decreased lung viral load and an increased IFN response. Overall, the data provided is convincing and supports the conclusion that GLDC represents a novel influenza virus susceptibility gene in cells and in vivo.

Minor comments:

1. I do not understand what the authors are attempting to show in Figure 1C.

In Fig 1C, we intend to show the connection of rs1755609 (the association SNP in H1N1 cohort) and rs2438409 (the association SNP in H7N9 cohort). The two study cohorts were applied to array genome-wide SNP analysis using different chips. The H1N1 association SNP rs1755609, also an expression quantitative locus of GLDC gene, was not incorporated in the chip used for genotyping H7N9 cohort samples. Thus, there is no rs1755609 genotyping data in H7N9 cohort. Based on linkage disequilibrium data of human genome, we selected a high linkage variant with rs1755609, i.e., SNP rs243840, and tested it in the H7N9 cohort. As shown in Table 2, rs2438409 variants are significant association with the susceptibility to H7N9 infection.

In Fig 1C, the numbers inside the boxes are r^2 measure. The grey box without number indicates the highest r^2 of 1.0. Figure 1C shows that r^2 measure in Chinese, Japanese and Caucasian are 0.97, 1.0, 1.0 respectively, indicating rs2438409 and rs1755609 (boxed in the new figure) are almost in complete linkage; namely, the alleles in these two polymorphic loci are inherited together, thus they can represent each other. A collective conclusion drawn from table 1, table 2 and Fig 1 is that, GLDC SNPs which are correlated to GLDC expression are associated with the susceptibility to H7N9 infection and severe H1N1 infections.

2. Has the rs1755609-A been identified in the many other GWAS examining influenza susceptibility? If not, how is this study different?

Unlike common diseases such as cancer or hypertension, severe influenza is relatively less prevalent. As far as we know, few GWAS has been performed to identify susceptibility gene to severe influenza. Most reported SNPs associated with severe influenza were generally based on candidate-gene approach (Everitt, Clare et al., 2012, Herrera-Ramos, Lopez-Rodriguez et al., 2014, Lee, Cao et al., 2017). Zuniga et al conducted a GWAS of ~50,000 SNPs to discover genetic variants associated with severe pneumonia in pandemic H1N1 infection (Zuniga, Buendia et al., 2011). Without any validation, the top five variants (the variants with the lowest P values) were reported solely based on genetic association. Our association SNPs have little connection with these five reported variants. The discrepancy is that we used a distinct strategy to select candidate association variants. With an ultimate goal of identifying important host gene(s) to affect disease outcome, rather than genetic markers associated with disease outcome, we integrated genetic association data with expression quantitative trait loci (eQTL) analysis. Namely, we do not select those SNPs with the lowest P values; instead we chose the SNPs which may not generate the highest association signal, but are correlated to differential expression of specific host genes. When the association SNP is transformed into a specific gene, we can proceed to validate the role of the gene for influenza infection in cellular and mouse study.

3. The rs34481144 eQTL for IFITM3 that is also associated influenza susceptibility should be discussed. Was this risk allele also included in your GWAS? Might these risk alleles be additive or are they found in distinct patients?

We thank the reviewer for sharing this important paper with us. rs34481144 was not incorporated into the chips, Genome-Wide Human SNP Array 6.0 (Affymetix) and HumanOmniZhongHua-8 BeadChip (Illumina) by which we genotyped the DNA samples of H1N1 and H7N9 cohorts respectively. In addition, according to the data of 1000 Genomes Project, rs34481144 A/G is polymorphic in some populations such as European (EUR) and American (AMR) (<https://www.ncbi.nlm.nih.gov/variation/tools/1000genomes/>). However, in Asian, including Southern Chinese Han (CHS) and Japanese (JPT), the frequency of risk allele A is 0%. Thus, despite of its association with severe influenza in patients of European descent (Allen, Randolph et al., 2017), rs34481144 may not associate with diseases in Asian.

4. Can the authors provide data or speculate as to how rs1755609 regulates GLDC expression?

The association SNP rs1755609 as an eQTL of GLDC is shown in multiple tissues (https://pubs.broadinstitute.org/mammals/haploreg/detail_v4.1.php?query=&id=rs1755609).

HaploReg, a powerful tool for functional annotation of non-coding variants, illustrates that rs1755609 and its high linkage SNPs overlap with promoter and enhancer in multiple cell types; the mutation results in changed motifs. Thus, the altered promoter and/or enhancer activity may modulate GLDC expression. Of course, more wet-lab based studies are required to dissect the molecular mechanism by which rs1755609 regulates GLDC expression.

5. Can the authors provide data or speculate as to how the pyrimidine biosynthesis pathway regulates type I IFN production? Are there other enzymes in this pathway that may be similarly involved?

We appreciate the reviewer for raising the interesting question, which has also intrigued us throughout this study. To date, various studies (Cheung, Lai et al., 2017, Lucas-Hourani, Dauzonne et al., 2013, Lucas-Hourani, Dauzonne et al., 2017), including ours, have convincingly established that inhibition of pyrimidine biosynthesis by various means can boost cellular interferon response. We expect that the mechanism underlying pyrimidine-depletion-mediated activation of innate immunity would be very complicated. Nevertheless, it is a very interesting question which deserves our further investigation in the future.

Referee #2 (Comments on Novelty/Model System for Author):

The interpretation of the data relies heavily, though not entirely, on the use of an ill-defined inhibitor. The specificity of that inhibitor for GLDC should be documented. Reference to ethical approval of the mouse challenge experiment and the ethical endpoints that were used to minimize animal suffering appear to be missing.

We used GLDC inhibitor in multiple experiments. However, except mouse experiment, the results of AOAA experiments were verified by GLDC siRNA depletion (Fig 2B, Fig 5A&5B) and GLDC overexpression (Fig 5C&5D), indicating the effects of AOAA in relevant experiments are specific. AOAA has been commonly used as a GLDC inhibitor in many studies since 1960s (Brunk & Rhodes, 1988). Glycine is the natural substrate of GLDC. AOAA is a substrate analog of GLDC with the amino (NH₂) group of glycine replaced by an aminooxy (ONH₂) group. Thus, AOAA was used to model the crystal structure of GLDC binding to substrate (Nakai, Nakagawa et al., 2005). Based on these reports, we utilized AOAA to showcase the effect of GLDC inhibition in cellular and mouse studies. We follow the reviewer's suggestion and describe the specificity of AOAA in the revised manuscript on page 12.

We have obtained the ethical approval for the mouse challenge experiment from our institute, Committee on the Use of Live Animals in Teaching and Research approved project number 4057-16 was specified in the original manuscript. The humane endpoint which had been specified in the approval is now incorporated in the revised manuscript on page 14.

Referee #2 (Remarks for Author):

The authors provide evidence based on an observational study in H1N1 and H7N9 patients with moderate or severe disease, that an intronic allele in the GLDC gene is associated with a more severe clinical outcome. Treatment of A549 cells with aminooxy-acetic acid combined with polyI,C transfection results in a higher type I IFN and ISG induction compared with control cells that were transfected with polyI,C only. Knock down of GLDC expression combined with polyI,C transfection had a similar effect. Treatment of A549 cells with AOAA reduced the replication of H1N1pdm and H7N9 infection, and was associated with increased type I IFN induction. Knock down of GLDC expression reduced H7N9 replication and overexpression of GLDC increased H7N9 replication in A549 cells. Finally, three consecutive intranasal AOAA treatments of mice partially protected the mice against disease following H1N1pdm challenge, which was associated with reduced virus replication and increased IFN α and ISG expression.

The link between the susceptibility to influenza and GLDC is novel and interesting. However, many of the results are based on the use of AOAA. The specificity and potential toxicity of this drug is poorly defined.

Major remarks:

1. Line 28. In contrast to the statement in the manuscript there is substantial evidence that pre-existing immunity has an important impact on the immune response and disease outcome following infection with pandemic H1N1 virus or H7N9 virus. This is presumably due to antigenic imprinting. See e.g. Gostic et al., Science 2016; reviewed in Cobey and Hensley, COV 2017). The statement should therefore be adapted.

We appreciate the reviewer's comment. Actually, we meant to state that no protective neutralization antibodies exist in human population since both viruses are novel reassortant strains and no one had been vaccinated against these strains prior to the time when we recruited and sampled the patients. The text has been revised accordingly on page 3.

2. Several susceptibility gene polymorphisms have been proposed to contribute to disease severity in influenza patients in the context of H1N1pdm or zoonotic infections with H7N9 viruses (IFITM3; Chen et al Sci Rep. 2016 May 9;6:25614. doi: 10.1038/srep25614.). The authors previously proposed that Galectin 1 was a susceptibility gene for H7N9. Now altered expression of GLDC is proposed as a susceptibility gene, mechanistically supposedly by its impact on pyrimidine biosynthesis, which in turn can suppress the innate antiviral response. There may thus be a more general effect on susceptibility to virus infection and GLDC expression or activation levels. The proposed link between GLDC and innate antiviral responses would be strengthened if the authors could extend the findings for another viral infection, for example VSV.

We thank the reviewer's constructive comment. Following the suggestion, we overexpressed GLDC or vector in A549 cells and examined the replication of MERS coronavirus (MERS-CoV), an emerging virus causing human respiratory infection (Zhou, Li et al., 2017). We demonstrated that GLDC overexpression significantly promoted the replication of MERS-CoV in the revised manuscript on page 7/8. The new data is submitted as new Fig 5F. The original Western Blot result is moved to Fig 5E.

3. It is unclear how specific an inhibitor of GLDC AOAA is. Also, in EV1, a concentration of 100 microM does not seem to result in a significant reduction in thymidine in A549 cells. The statement in line 122 (GLDC inhibition could suppress pyrimidine biosynthesis) is therefore not supported by the data. The specificity of AOAA should be documented with GLDC knock down or -deficient cells before conclusions based on experiments with AOAA can be made. GLDC knock down cells were generated by the authors. The statement in line 125 (GLDC inhibition triggered activation of type I IFNs) is not correct. Without polyI,C transfection there appears to be no effect on type IFN or ISG induction.

We agree with the reviewer about the AOAA-mediated suppression of pyrimidine biosynthesis. GLDC regulation of pyrimidine biosynthesis and the mechanism behind have been well-established (Zhang, Shyh-Chang et al., 2012). GLDC-catalyzed reaction converts glycine into CH₂-THF (Kume, Koyata et al., 1991); the latter contains a methylene group that fuels de novo thymidine synthesis (Tibbetts & Appling, 2010). In our experiment, although AOAA showed a dose-dependent inhibitory effect, 100uM AOAA reduced the abundance of thymidine with a borderline significance (P = 0.061). We used LC-MS/MS, a very sensitive approach, to detect the amount of cellular thymidine, intending to verify the previous reports. Anyway, we take the reviewer's suggestion, tone down and state that "AOAA appeared to diminish the cellular pool of thymidine nucleotide in a dose-dependent manner" (page 5).

GLDC is a pyridoxal 5'-phosphate (PLP) dependent enzyme, catalyzing the degradation of glycine. AOAA occupies the substrate-binding site and inhibit GLDC activity (Nakai et al., 2005). AOAA has been used for GLDC inhibition in many studies since 1960s. We understand the reviewer suggests us to evaluate the effect of AOAA for boosting antiviral response and suppressing viral growth in GLDC siRNA depleted cells. Theoretically, AOAA-mediated effect would be diminished in GLDC siRNA depleted cells compared with the control cells. Although it sounds like a good experiment to show the specificity of AOAA, we are afraid that it may not work. As shown in Fig 3B, AOAA reduced H7N9 viral titer by 2 log units, whilst the reduction of viral titer was around 3 folds in GLDC siRNA depleted cells (Fig 5A). This is conceivable since the GLDC chemical inhibition by a small molecule AOAA is definitely more potent than siRNA depletion. Thus, the contribution of siRNA-mediated GLDC depletion in antiviral response and viral growth would be easily masked by

AOAA-mediated effect. We added a paragraph at the end of discussion section, stating that more verification of the specificity of AOAA is required.

In line 125, we stated that "GLDC inhibition triggered activation of type I IFNs and ISGs required the PRR signalling", which is consistent with the reviewer's comment.

4. Related to point 3: do DD264 or brequinar still have an anti-influenza virus effect in GLDC knock down cells?

Brequinar is a chemical inhibitor of DHODH, an important enzyme for pyrimidine biosynthesis; and DD264 was identified as a broad-spectrum antiviral, probably targeting DHODH (Lucas-Hourani et al., 2013). Based on our data, GLDC inhibition may converge with DD264 and brequinar at pyrimidine metabolism, thus shares similar antiviral mechanism with these two compounds. We believe that these two compounds may act similarly to AOAA; their high potency to inhibit pyrimidine biosynthesis may mask the downstream effects of siRNA-mediated GLDC depletion.

5. The anti-H1N1 and H7N9 effect of AOAA seems clear in infected A459 cells as are the effects on IFN- α and - β , and ISG expression. Please clarify that in these experiments AOAA was only applied to the cells before infection (pretreatment).

We thank the reviewer's comment. In fact, we pretreated the cells with AOAA for one hour prior to virus inoculation. After inoculation, AOAA was supplemented in the culture medium till the end of experiment. we have revised the manuscript on page 14. We apologize for the obscure description in the initial manuscript.

6. The mouse experiment documents limited protection of intranasal AOAA treatment of H1N1 pdm infected BALB/c mice. With 10 mice per group it is unclear what 12.5% survival means. In addition, the experiment should be repeated with H1N1 pdm or with H7N9 virus challenge. The authors should also provide data on the pharmacokinetics and possible toxicity of AOAA in vivo. GLDC deficiency in humans is associated with severe developmental defects.

We thank the reviewer for identifying the mistake in our manuscript. The survival data was the pooled results of two experiments. We checked the raw data and found that we actually had 9 mice for each group; All AOAA- and Zanamivir-treated mice survived, one PBS-treated mouse survived the experiment. Thus, the survival rate of PBS mice was 11.1%. we apologize for the wrong calculation in the first place.

Although mice are the most commonly used animal for influenza research, they are distinct from humans in many aspects. Most human influenza viruses cannot effectively infect Balb/c mice and cause fatal infection. On the other hand, mouse survival is a golden standard for most influenza mouse studies. Thus, the influenza viruses infective to humans, including H1N1pdm virus, have to be passaged in Balb/c mice multiple times in order to obtain mouse-adapted strain which is able to cause lethal infection in mice. Then the mouse-adapted analogue is used in mouse experiments. This was exactly what we have done. We previously obtained a mouse-adapted H1N1pdm strain (Zheng, Chan et al., 2010) and we used this strain, instead of the original H1N1pdm, in this study. As shown in the reference, after inoculation of very high titer (10^6 pfu) of wild-type H1N1pdm, the mouse survival rate was 100%. This issue has been specified in the original manuscript on page 14 line 367-369.

We indeed tested the toxicity of AOAA in A549 cells and Balb/c mice, which is now submitted as Fig EV3 and Fig EV5 respectively. In general, AOAA is not cytotoxic at the concentrations used for in vitro and in vivo studies.

Other remarks:

1. Panel 4E fits better in figure 2 where the GLDC KD cells are described.

In Panel 4E (Panel 5E in the re-submission), we monitored GLDC knockdown in the course of infection, i.e., 24 and 48 hours post infection, intending to verify GLDC depletion throughout the infection experiment and to exclude the possibility that virus infection may boost the expression of

depleted GLDC. In Fig 2, no infection was involved. The knockdown effect of GLDC siRNA has been consistently verified by Western blot although we don't show the Western blot result in Fig 2.

2. GLDC expression levels are high in liver, kidney and placenta but low in the lung (Kure et al., J Hum Genet. 2001;46(7):378-84). How do the authors interpret their findings in the light of that report?

We thank the reviewer's comment. We also noticed that GLDC expression level in lung is lower than that in other organs such as liver and kidney. Nevertheless, we hope the reviewer agree with us that functionally important genes for disease pathogenesis are not necessarily the highly expressed ones. Although GLDC is not highly expressed in lung, rs1755609 is a strong eQTL in lung as shown in Fig 1B.

3. Line 254: "the outperformance...": compared to what? Certainly not to zanamivir administration.

We meant that AOAA in mouse experiment is surprisingly better than that in cellular experiments of A549 cells. The "outperformance of ..." was followed by "We further demonstrated that in the H1N1-infected Balb/c mice, AOAA-mediated amplification of antiviral response and suppression of viral growth are even more prominent than its in vitro effect in A549 cells."

4. Please specify whether the mice were sedated or not for the intranasal inoculations.

We followed the protocol approved by Lab Animal Unit of our institute, an animal center with AAALAC accreditation. In the approved protocol, mice are anesthetized by intraperitoneal injection of ketamine/xylazine cocktail which contains 70-100mg/kg ketamine and 10-20mg/kg xylazine prior to intranasal administration and inoculation. We have incorporated these details in the manuscript accordingly on page 14.

5. Line 433: antiviral should be replaced with anti-influenza A.

The manuscript has been revised accordingly on page 17.

Referee #3 (Comments on Novelty/Model System for Author):

In the animal model, would the authors please explain the criteria for evaluating illness and death after infection that is presented in Figure 4B and EV4. As presented in EV4, the weight loss observed in the PBS group is moderate for an influenza virus infection model, with the average weight loss at about 12% and low points in the weight loss curves observed at days 4 and 9 after infection. Do the authors have any explanation for the bimodal weight loss in these animals that demonstrates recovery at Day 6 followed by weight loss at Day 9? This is especially interesting when aligning the survival curves (Figure 4B) with weight loss because deaths occur on Days 5, 7, and 10 after infection. This observed weight loss can explain the deaths at Day 5 (with low weights at Day 4), but do not explain the deaths at Day 8 when weights return to nearly 100% at Day 7. The loss of weight at Day 9 corresponds with the death at Day 10. Two things are worth noting from the weight: survival comparison. First, the weight loss after infection is not consistent with the weight loss observed in other lethal models of influenza virus infection where a consistent progression toward 25-30% weight loss is observed. Second, a major loss of animals due to infection is associated with increases in group body weight (Day 7/8), while a minimal number of deaths is associated with a decrease in body weight (Day 9/10). If the authors could clarify how this weight loss is associated with the survival observed, and specifically comment on the euthanasia criteria for these animals, it would be greatly appreciated.

We thank the reviewer for identifying the inconsistent data in survival rate and body weight loss. we checked the raw data and found the wrong calculation in the initial manuscript. Instead of 10 mice in each group, we actually had three groups of 9 mice treated with AOAA, Zanamivir and PBS in that experiment. All AOAA- and Zanamivir-treated mice survived the virus challenge; whilst only one PBS-treated mouse was alive at the end of the experiment, with a final survival rate of 11.1%. According to the approved protocol, we should monitor mouse body weight loss, the mice with excessive weight loss >25% are euthanized to minimize mouse suffering. Thus, the mice with weight

loss >25% were excluded for body weight calculation. However, we forgot to synchronize with the survival data, which caused the discrepancy between body weight loss and survival rate. We replotted the survival data and submitted a new version of Fig 6B. We apologize for wasting your time to interpret our wrongly-presented data.

We agree with the reviewer that PBS-treated mice showed an unusual bimodal weight loss. As mentioned above, there were 9 mice mock treated with PBS at the beginning of the experiment; 4 and 2 mice remained at day 5 and day 6 respectively. At day 5, 2 mice (No.3 and 4) lost body weight around 21%, which decreased the mean body weight substantially. After they died at day 6, the mean body weight of two remaining mice rebounded until one of them (No. 2 mouse) eventually lose weight and died on day 10. The raw data of mouse body weight is presented below for your perusal. As shown in our raw data, all the mice dying from the infection lost body weight progressively. However, No.2 mouse became sick relatively later than the other mice, which caused the unusual curve of mean body weight.

Body weight change (% of original weight) of 9 PBS-treated mice.

Day post infection	Mouse No.								
	1	2	3	4	5	6	7	8	9
-1	100.0	100.0	100.0	100.0	100.0	100.0	100.0	100.0	100.0
0	102.1	103.6	104.7	103.8	102.6	96.9	90.1	91.9	83.3
1	95.8	98.5	98.8	101.6	95.4	97.5	91.1	94.4	83.3
2	93.8	97.4	100.0	100.0	93.9	95.9	91.1	92.4	82.8
3	83.9	95.9	88.8	93.5	85.1	101.0	82.2	86.3	79.8
4	78.7	98.0	82.9	87.5	79.0	97.5		77.7	
5		98.5	79.4	79.9		98.0			
6		94.4				96.4			
7		92.3				93.4			
8		80.5				95.4			
9		77.4				88.3			
10						88.3			
11						92.9			
12						94.4			
13						94.4			
14						94.4			

Referee #3 (Remarks for Author):

Comments for the authors of EMBO Molecular Medicine manuscript number EMM-2018-09528: The authors of the EMBO Molecular Medicine manuscript "Identification and characterization of GLDC as host susceptibility gene to severe influenza", present their method for demonstrating that glycine decarboxylase (GLDC) is associated with severe influenza infections. They begin this study by identifying GLDC using a genome-wide association study in patient populations that presented with severe infections caused by either H7N9 or pandemic H1N1 viruses. Genetic evaluation of these patients showed an association between severe a GLDC SNP and severe influenza infections. The authors then tested the contribution of GLDC to severe influenza infection by initially testing the response of influenza virus-infected cells to treatment with the GLDC inhibitor (aminooxy)acetic acid (AOAA) in A549 cells. Specifically, using a luciferase system, the authors show that type I IFN expression is increased in the presence of the GLDC inhibitor AOAA, indicating that GLDC limits anti-viral activity. These findings are corroborated using siRNA studies that show increased type I IFN expression when GLDC is depleted.

Having established an association between GLDC expression and reduced anti-viral immunity, modulated by type I IFN expression, the authors further show that AOAA treatment reduces viral infection of A549 cells, an effect that is again associated with increased expression of type I IFNs and subsets of IFN-stimulated genes (ISGs). Interestingly, the increase in ISGs is both more robust and broad after infection with H7N9 when compared with H1N1, indicating potential viral gene

contributions to the phenotypes observed. The authors then use both siRNA depletion of GLDC and GLDC overexpression to show that GLDC expression directly affects virus replication, type I IFN expression, and ISG levels after infection. Finally, in a mouse infection model the authors show that AOAA is as effective as zanamivir at preventing death after influenza virus infection, an outcome that is associated with increased virus levels in lungs, IFN-alpha expression, ISG expression, and detection of virus nucleoprotein in the lung.

This manuscript does an excellent job of progressing the study from an initial characterization of a human gene associated with severe influenza virus infection to demonstrating that inhibition of this gene activity can directly improve outcomes after influenza virus infection. The authors show that the host type I IFN response, in particular ISGs, are important for the outcomes observed both through chemical and genetic inhibition of GLDC activity in cell lines and mice. This progression from a human gene target through demonstration that specific inhibition of gene function can directly influence the fate of an infection is a logical and complete progression of this gene from initial characterization to demonstrated gene function. While the information presented is of interest, there are a few issues with the presentation of the information and discussion of the findings that I would like the authors to consider as they evaluate the presentation of these findings.

General Comments:

1. One major concern with the data presented is the eventual utilization of a therapeutic that targets a gene product that needs to be inhibited in order to demonstrate an effect. Treatment of individuals with this product will need to overcome variations in gene expression within the human population that may greatly influence dosing and overall effectiveness. The most direct demonstration of effectiveness in a therapeutic setting is the murine study where AOAA is used to prevent death after infection, but the initial infection is still associated with weight loss and recovery. Furthermore, the fact that AOAA is administered on days -1, 0, and +1 after infection brings into question the potential application of this therapeutic. How do the authors propose therapeutic treatment after influenza virus infection?

We thank the reviewer's comment. The results of genetic association and eQTL analysis implicate that lower GLDC expression may confer protection from severe influenza infection, from which we generated the hypothesis. The downstream cellular and mouse experiments test the hypothesis and establish that GLDC inhibition boosted antiviral response, suppressed viral growth and ameliorated disease outcome. The potential application of GLDC inhibition for prevention or treatment of severe influenza is based on the conclusion of cellular characterization and mouse experiment. In this regard, the variable GLDC expression in human population should not be a serious consideration.

Inhibiting a gene product for therapeutic purpose is quite common in clinical medicine. Many inhibitors against host cellular kinases have been approved by FDA for cancer therapy (Bhullar, Lagaron et al., 2018). In this study, our purpose of using AOAA in mouse study is to demonstrate the role of GLDC for development of severe influenza, although our data implicated the potential therapeutic application of GLDC inhibition.

Our design of administering AOAA on day -1, 0 and +1 after virus inoculation aimed to maximize the effect of AOAA in the course of influenza infection. We may consider to explore the possibility of using GLDC inhibitors for therapeutic purpose, e.g., evaluating the effect of AOAA after administration at day 1, day 2 and day 3 post infection. Interestingly, due to the importance of GLDC for tumorigenesis and tumor growth, scientists are now screening GLDC inhibitors for prevention and treatment of malignant tumors (<https://patents.google.com/patent/US20130296188A1/en>). The new generation of GLDC inhibitors would be a good example of killing two birds with one stone, i.e. repurposing anticancer drugs for treatment of severe influenza.

2. In the animal model, would the authors please explain the criteria for evaluating illness and death after infection that is presented in Figure 4B and EV4. As presented in EV4, the weight loss observed in the PBS group is moderate for an influenza virus infection model, with the average weight loss at about 12% and low points in the weight loss curves observed at days 4 and 9 after infection. Do the authors have any explanation for the bimodal weight loss in these animals that demonstrates recovery at Day 6 followed by weight loss at Day 9? This is especially interesting when aligning the survival curves (Figure 4B) with weight loss because deaths occur on Days 5, 7,

and 10 after infection. This observed weight loss can explain the deaths at Day 5 (with low weights at Day 4), but do not explain the deaths at Day 8 when weights return to nearly 100% at Day 7. The loss of weight at Day 9 corresponds with the death at Day 10. Two things are worth noting from the weight: survival comparison. First, the weight loss after infection is not consistent with the weight loss observed in other lethal models of influenza virus infection where a consistent progression toward 25-30% weight loss is observed. Second, a major loss of animals due to infection is associated with increases in group body weight (Day 7/8), while a minimal number of deaths is associated with a decrease in body weight (Day 9/10). If the authors could clarify how this weight loss is associated with the survival observed, and specifically comment on the euthanasia criteria for these animals, it would be greatly appreciated.

We addressed these inquiries in the text above.

3. Would the authors be willing to comment on the potential application of these findings and the therapeutic approach toward future treatments in humans. In particular, did the patients from which the genetic information was first derived have co-infections or secondary bacterial infections? How does GLDC influence secondary bacterial infections? Would AOAA therapy be expected to affect progression to a secondary bacterial infection? Would this effect be prevention or enhancement of this secondary infection?

We have addressed the potential therapeutic application in the text above. The issue of bacterial co-infection is only relevant in H1N1 cohort where we had criteria to define a severe versus mild patient. H1N1pdm infection was confirmed by either positive results of RT-PCR or virus culture of respiratory tract specimens. Patients of severe infection were defined as those who required oxygen supplementation or admitted to the intensive care unit or succumbed to the infection (Zhou, To et al., 2012). Thus, we did not select severe patients on basis of the incidence of bacterial co-infection. However, according to previous clinical studies, bacterial co-infection is frequent in patients with influenza infection, with incidence rate around 20–30% (Klein, Monteforte et al., 2016, Rice, Rubinson et al., 2012). In addition, co-infection is associated with higher mortality rate than primary viral infection. Thus, bacterial co-infection probably occurred in many of our severe patients. we believe that, as a potential therapeutic option, GLDC inhibition would prevent the incidence of bacterial co-infection due to the enhanced innate immunity.

4. The genetic observation made does not seem to associate disease severity with a loss of gene expression, a tool that was used to demonstrate the role of GLDC in the cell culture studies. In the individuals that have different SNPs for GLDC, are the levels of protein expression affected? What are the subsequent type I IFN responses in these individuals? Since these are type I IFNs, their baseline levels may be increased when compared to individuals with high levels of GLDC expression. Similarly, does ISG expression differ in these individuals? Finally, can AOAA be used clinically, and does it have any potential alternative effects that could be detrimental to the progression of a virus infection and/or a secondary bacterial infection?

Based on the genetic association and eQTL analysis, we demonstrated that alleles with higher GLDC-expressing alleles are associated with higher risk to H7N9 infection and severe H1N1 infection. The risk alleles are not loss-of-function variants. In addition, eQTL data delineate the genome-wide correlation of genetic variation and mRNA expression of host genes. A genetic variant correlated to higher mRNA expression of a cellular gene may result in higher protein expression. Anyway, the reviewer raised an important question. we are planning to verify in an appropriate system that cells carrying different alleles show differential levels of GLDC protein expression.

We believe that GLDC expression level may not affect the baseline expression of type I IFNs and ISGs. GLDC inhibition or depletion can boost the induction of type I IFNs and ISGs only when PRRs are stimulated or upon viral infection. Namely, the genetically-regulated differential GLDC expression in humans may not affect baseline levels of type I IFNs and ISGs; instead it can affect the magnitude of IFN and ISG induction upon viral infection.

As mentioned above, we believe GLDC inhibition via inhibitors or antagonists could be developed as prevention or therapeutics against viral infections due to its effect to boost host antiviral response.

5. Overall, the decrease in virus titers in the lungs of mice (Figure 4C) was approximately 1-2 logs with high levels still detected at Day 5 post-infection. Could the authors comment on the potential impact of a lingering virus infection after AOAA treatment.

As shown in Fig 6C, GLDC inhibition boosted antiviral response and dampened viral replication in mice. The enhanced host defense can facilitate viral clearance, but unable to shut off viral growth. We chose day 3 and day 5 for detection since these are usually the time points with the highest viral growth. After day 5, most infected mice either deteriorate and eventually succumb to infection, or resolve the virus and recover. As an acute infection, influenza virus infection is not related to any virus lingering. The mouse survival data (Figure 6B) shows clearly that all AOAA-treated mice recovered from the infection which killed 89% of PBS-treated mice.

Specific comments

1. While understandable at the level of data presentation and evaluation, the manuscript would benefit from substantial editing for sentence and paragraph structure throughout.

We appreciate the reviewer's suggestion and have tried our best to polish our writing.

Reference:

- Allen EK, Randolph AG, Bhangale T, Dogra P, Ohlson M, Oshansky CM, Zamora AE, Shannon JP, Finkelstein D, Dressen A, DeVincenzo J, Caniza M, Youngblood B, Rosenberger CM, Thomas PG (2017) SNP-mediated disruption of CTCF binding at the IFITM3 promoter is associated with risk of severe influenza in humans. *Nat Med* 23: 975-983
- Bhullar KS, Lagaron NO, McGowan EM, Parmar I, Jha A, Hubbard BP, Rupasinghe HPV (2018) Kinase-targeted cancer therapies: progress, challenges and future directions. *Mol Cancer* 17: 48
- Brunk DG, Rhodes D (1988) Amino Acid Metabolism of Lemna minor L. : III. Responses to Aminoxyacetate. *Plant Physiol* 87: 447-53
- Cheung NN, Lai KK, Dai J, Kok KH, Chen H, Chan KH, Yuen KY, Kao RYT (2017) Broad-spectrum inhibition of common respiratory RNA viruses by a pyrimidine synthesis inhibitor with involvement of the host antiviral response. *J Gen Virol* 98: 946-954
- Everitt AR, Clare S, Pertel T, John SP, Wash RS, Smith SE, Chin CR, Feeley EM, Sims JS, Adams DJ, Wise HM, Kane L, Goulding D, Digard P, Anttila V, Baillie JK, Walsh TS, Hume DA, Palotie A, Xue Y et al. (2012) IFITM3 restricts the morbidity and mortality associated with influenza. *Nature* 484: 519-23
- Herrera-Ramos E, Lopez-Rodriguez M, Ruiz-Hernandez JJ, Horcajada JP, Borderias L, Lerma E, Blanquer J, Perez-Gonzalez MC, Garcia-Laorden MI, Florido Y, Mas-Bosch V, Montero M, Ferrer JM, Sorli L, Vilaplana C, Rajas O, Briones M, Aspa J, Lopez-Granados E, Sole-Violan J et al. (2014) Surfactant protein A genetic variants associate with severe respiratory insufficiency in pandemic influenza A virus infection. *Crit Care* 18: R127
- Klein EY, Monteforte B, Gupta A, Jiang W, May L, Hsieh YH, Dugas A (2016) The frequency of influenza and bacterial coinfection: a systematic review and meta-analysis. *Influenza and other respiratory viruses* 10: 394-403
- Kume A, Koyata H, Sakakibara T, Ishiguro Y, Kure S, Hiraga K (1991) The glycine cleavage system. Molecular cloning of the chicken and human glycine decarboxylase cDNAs and some characteristics involved in the deduced protein structures. *J Biol Chem* 266: 3323-9
- Lee N, Cao B, Ke C, Lu H, Hu Y, Tam CHT, Ma RCW, Guan D, Zhu Z, Li H, Lin M, Wong RYK, Yung IMH, Hung TN, Kwok K, Horby P, Hui DSC, Chan MCW, Chan PKS (2017) IFITM3, TLR3, and CD55 Gene SNPs and Cumulative Genetic Risks for Severe Outcomes in Chinese Patients With H7N9/H1N1pdm09 Influenza. *J Infect Dis* 216: 97-104
- Lucas-Hourani M, Dauzonne D, Jorda P, Cousin G, Lupan A, Helynck O, Caignard G, Janvier G, Andre-Leroux G, Khair S, Escriou N, Despres P, Jacob Y, Munier-Lehmann H, Tangy F, Vidalain PO (2013) Inhibition of pyrimidine biosynthesis pathway suppresses viral growth through innate immunity. *PLoS Pathog* 9: e1003678
- Lucas-Hourani M, Dauzonne D, Munier-Lehmann H, Khair S, Nisole S, Dairou J, Helynck O, Afonso PV, Tangy F, Vidalain PO (2017) Original Chemical Series of Pyrimidine Biosynthesis Inhibitors That Boost the Antiviral Interferon Response. *Antimicrob Agents Chemother* 61
- Nakai T, Nakagawa N, Maoka N, Masui R, Kuramitsu S, Kamiya N (2005) Structure of P-protein of the glycine cleavage system: implications for nonketotic hyperglycinemia. *EMBO J* 24: 1523-36

Rice TW, Rubinson L, Uyeki TM, Vaughn FL, John BB, Miller RR, 3rd, Higgs E, Randolph AG, Smoot BE, Thompson BT, Network NA (2012) Critical illness from 2009 pandemic influenza A virus and bacterial coinfection in the United States. *Crit Care Med* 40: 1487-98

Tibbetts AS, Appling DR (2010) Compartmentalization of Mammalian folate-mediated one-carbon metabolism. *Annu Rev Nutr* 30: 57-81

Zhang WC, Shyh-Chang N, Yang H, Rai A, Umashankar S, Ma S, Soh BS, Sun LL, Tai BC, Nga ME, Bhakoo KK, Jayapal SR, Nichane M, Yu Q, Ahmed DA, Tan C, Sing WP, Tam J, Thirugananam A, Noghabi MS et al. (2012) Glycine decarboxylase activity drives non-small cell lung cancer tumor-initiating cells and tumorigenesis. *Cell* 148: 259-72

Zheng B, Chan KH, Zhang AJ, Zhou J, Chan CC, Poon VK, Zhang K, Leung VH, Jin DY, Woo PC, Chan JF, To KK, Chen H, Yuen KY (2010) D225G mutation in hemagglutinin of pandemic influenza H1N1 (2009) virus enhances virulence in mice. *Exp Biol Med (Maywood)* 235: 981-8

Zhou J, Li C, Zhao G, Chu H, Wang D, Yan HH, Poon VK, Wen L, Wong BH, Zhao X, Chiu MC, Yang D, Wang Y, Au-Yeung RKH, Chan IH, Sun S, Chan JF, To KK, Memish ZA, Corman VM et al. (2017) Human intestinal tract serves as an alternative infection route for Middle East respiratory syndrome coronavirus. *Sci Adv* 3: eaao4966

Zhou J, To KK, Dong H, Cheng ZS, Lau CC, Poon VK, Fan YH, Song YQ, Tse H, Chan KH, Zheng BJ, Zhao GP, Yuen KY (2012) A functional variation in CD55 increases the severity of 2009 pandemic H1N1 influenza A virus infection. *J Infect Dis* 206: 495-503

Zuniga J, Buendia I, Zhao Y, Jimenez L, Torres D, Romo J, Ramirez G, Cruz A, Vargas-Alarcon G, Sheu CC, Chen F, Su L, Tager AM, Pardo A, Selman M, Christiani DC (2011) Genetic variants associated with severe pneumonia in A/H1N1 influenza infection. *Eur Respir J*

2nd Editorial Decision

26 September 2018

Thank you for the submission of your revised manuscript to EMBO Molecular Medicine. We have now received the enclosed reports from the referees that were asked to re-assess it. As you will see the reviewers are globally supportive. Still, referees 1 and 2 request additional text changes and importantly, referee 1 really would like to see additional in vitro experiments to strengthen the results, which I would strongly advise you to perform.

Please submit your revised manuscript as soon as possible.

***** Reviewer's comments *****

Referee #1 (Remarks for Author):

My original comments were largely requests for clarification, which the authors have mostly provided. The question raised in comment #5 asking the authors to speculate as to how the pyrimidine biosynthesis pathway regulates type I IFN production was not answered. I still think that a possible mechanism should be discussed. Relating to this, I also agreed with several comments of the other reviewers and observe that many of these comments were also not addressed. Validation of the specificity of AOAA was not provided, nor was testing of other pyrimidine biosynthesis inhibitors. These simple in vitro experiments would provide much greater confidence in the conclusions of the manuscript. Overall, I find the authors to have been largely unresponsive to the reviewer critiques.

Referee #2 (Comments on Novelty/Model System for Author):

Overall, the experiments allow to conclude that GLDC can negatively regulate type I IFN responses and therefore can modulate susceptibility to influenza virus infection.

Referee #2 (Remarks for Author):

Line 26: "Thus, pre-existing immunity due to vaccination and prior exposure cannot account for the susceptibility to infections of these viruses" goes against Gostic et al. who stated "Potent protection

against H5N1 and H7N9 influenza via childhood hemagglutinin imprinting". In other words, the influenza A virus infection encountered as a child correlates with live long protection from novel influenza subtypes in the same phylogenetic group. Thus a H3N2 infection (prior exposure) in childhood confers a degree of protection against later H7N9 infection. Please adapt the statement.

The specificity of aminooxyacetic acid is not clear. The cited Cell paper by Zhang et al (2012) on glycine decarboxylase in cancer does not refer to this compound. Nagai et al indeed show the crystal structure of the P protein of GLDC in complex with AOAA. Wybenga et al. (JBC 2012) reported that aminooxyacetic acid is an inhibitor of a bacterial aminotransferase. The molecule appears to be a more general aminotransferase inhibitor. Therefore, I am pleased to read in the discussion that a note of caution on the specificity of AOAA is warranted.

I wish to thank the authors for having assessed MERS CoV in the context of GLDC overexpression.

Referee #3 (Remarks for Author):

I thank the authors for addressing my initial comments. I have no further comments regarding data presentation and interpretation.

2nd Revision - authors' response

3 November 2018

Referee #1

My original comments were largely requests for clarification, which the authors have mostly provided. The question raised in comment #5 asking the authors to speculate as to how the pyrimidine biosynthesis pathway regulates type I IFN production was not answered. I still think that a possible mechanism should be discussed. Relating to this, I also agreed with several comments of the other reviewers and observe that many of these comments were also not addressed. Validation of the specificity of AOAA was not provided, nor was testing of other pyrimidine biosynthesis inhibitors. These simple in vitro experiments would provide much greater confidence in the conclusions of the manuscript. Overall, I find the authors to have been largely unresponsive to the reviewer critiques.

We are very sorry for not appropriately answering the #5 question in the first review. Your question was "Can the authors provide data or speculate as to how the pyrimidine biosynthesis pathway regulates type I IFN production? Are there other enzymes in this pathway that may be similarly involved?"

As I mentioned, many scientists including us have been interested in the mechanism for the link between pyrimidine deprivation and amplified antiviral immunity, which has almost become a consensus recognition. After more extensive search, the only piece of supportive evidence we obtained so far is that a pyrimidine inhibitor identified as a broad-spectrum antiviral was found to activate RIG-I, an essential cellular sensor to mount innate immune response in virus infection, especially influenza virus infection. This has been discussed in the revised manuscript one page 12-13.

We agree with you that testing other pyrimidine inhibitor would be conducive to validate the specificity of AOAA after you emphasize herein. DHODH is an essential enzyme in pyrimidine biosynthesis pathway. Lots of experimental evidence demonstrate that inhibition of DHODH can boost antiviral immunity and suppress viral growth. Brequinar is well-characterized inhibitor of DHODH. Thus, we used brequinar to replace AOAA and performed the same H7N9 infection experiment as shown in Fig 3 and 4. The results of brequinar experiment, now submitted as new Fig EV3, basically recapitulated the findings of AOAA treatment. Brequinar treatment also upregulated IFN α and ISGs and significantly reduced H7N9 replication. Thus, DHODH is another enzyme involved in pyrimidine biosynthesis pathway that acts similarly as GLDC to regulate type I IFN production. We present the new data and revised the manuscript on page 3 and page 7.

Referee #2 (Remarks for Author):

Line 26: "Thus, pre-existing immunity due to vaccination and prior exposure cannot account for the susceptibility to infections of these viruses" goes against Gostic et al. who stated "Potent protection against H5N1 and H7N9 influenza via childhood hemagglutinin imprinting". In other words, the influenza A virus infection encountered as a child correlates with live long protection from novel influenza subtypes in the same phylogenetic group. Thus a H3N2 infection (prior exposure) in childhood confers a degree of protection against later H7N9 infection. Please adapt the statement.

We appreciate your comments. A statement is now added in the revised manuscript on page 9.

Corresponding Author Name: Kwok-Yung YUEN
Journal Submitted to: EMBO Molecular Medicine
Manuscript Number: EMM-2018-09528